# On the Generalization and Approximation Capacities of Neural Controlled Differential Equations

**Linus Bleistein**
Inria Paris, UEVE

**Agathe Guilloux**
Inria Paris

## Abstract

Neural Controlled Differential Equations (NCDEs) are a state-of-the-art tool for supervised learning with irregularly sampled time series (Kidger et al., 2020). In this article, we provide the first statistical analysis of their performance by merging the rich theory of controlled differential equations (CDE) and Lipschitz-based measures of the complexity of deep neural nets. Our first result is a generalization bound for this class of predictors that depends on the regularity of the time series data. In a second time, we leverage the continuity of the flow of CDEs to provide a detailed analysis of both the sampling-induced bias and the approximation bias. Regarding this last result, we show how classical approximation results on neural nets may transfer to NCDEs. Our theoretical results are validated through a series of experiments.

## 1 Introduction

Time series are ubiquitous in many domains such as finance, agriculture, economics and healthcare. A common set of tasks consists in predicting an outcome $y \in \mathcal{Y}$, such as a scalar or a label, from a time-evolving set of features. This problem has been addressed with a great variety of methods, ranging from Vector Auto-Regressive (VAR) models (Hamilton, 2020) and Gaussian processes (Roberts et al., 2013), to deep models such as Recurrent Neural Networks (RNN, Elman, 1991) and Long-Short-Term-Memory Networks (LSTM, Hochreiter & Schmidhuber, 1996).

**Irregular Time Series.** In real-world scenarios, time series are often irregularly sampled (when data collection is difficult or expensive for instance). Mathematically, for a time series $\mathbf{x} = (\mathbf{x}_{t_0}, \ldots, \mathbf{x}_{t_K}) \in \mathbb{R}^{d \times (K+1)}$, this means that $\Delta_j := t_j - t_{j-1}$ may vary with $j = 1, \ldots, K$. We are interested in understanding the effect of coarseness of sampling on the learning performance. Intuitively, coarser sampling makes learning harder: bigger gaps between sampling times increase the probability of missing important events - say, a spike in the data - that help predict the outcome. Formally, it is natural to consider that the time series $\mathbf{x}$ is a degraded version - through subsampling - of an unobserved underlying path $x = (x_t)_{t \in [0,1]}$ which, in turn, determines the outcome $y \in \mathcal{Y} \subset \mathbb{R}$. On a high level perspective, we have that $y = F(x) + \varepsilon$, where $F$ is an operator that maps the space of paths $\{x : [0,1] \to \mathbb{R}^d\}$ to $\mathcal{Y}$, and $\varepsilon$ is a noise term. For instance, in healthcare, the value of a vital of interest of a patient is determined by the continuous and unobserved trajectory of her biomarkers, rather than by the discrete measurements made by a physician.

**Models for Irregular Time Series.** While many models and approaches have been proposed for learning from irregular time series (see Section 2), a quite fruitful point of view is to view the irregular data and the outcome as being linked through a dynamical system. In this article, we focus on Neural Controlled Differential Equations (NCDE), which take this point of view. NCDEs learn a time-dependant vector field, continuously evolving with the streamed data, that steers a latent state. The terminal value of this latent state is then eventually used for prediction - as one would for instance do with RNNs.

**Our setup.** We restrict our attention to a supervised learning setup in which sampling times are arbitrarily spaced. Consider a sample $\{(y^i, \mathbf{x}^{D,i})\}$, where $y^i \in \mathcal{Y}$ is the label of a time series

$\mathbf{x}^{D,i} \in \mathbb{R}^{d \times (K+1)}$ discretized on an arbitrary grid $D = \{t_0, \ldots, t_K\}$. We will always require that this sampling grid is constant across individuals, and assume that $t_0 = 0$ and $t_K = 1$. We consider a predictor $\hat{f}^D$ obtained by empirical risk minimization (ERM) on this sample.

While NCDE have been proven to yield excellent performance on supervised learning tasks (Kidger et al., 2020; Morrill et al., 2021a; Qian et al., 2021; Seedat et al., 2022; Norcliffe et al., 2023), the literature on their theoretical properties is still sparse. Remarkably, Kidger (2022) proved that NCDEs are universal approximators i.e. that given a wide enough neural vector field, a NCDE can approximate any continuous function mapping the space of paths to a target space. This result hence transfers a classical result on functional approximation by neural networks to NCDEs. In this article, we aim at filling a theoretical gap on the learning capacities of NCDEs by adressing three critical questions.

1. How do NCDE *generalize* i.e. how well does $\hat{f}^D$ perform on unseen data drawn from the same distribution that the training data ? How strong is the curse of dimensionality for NCDEs ?

2. How well can NCDE *approximate* general controlled dynamical systems ?

3. How does the *irregular sampling* affect generalization and approximation ? Put otherwise, how does the predictor $\hat{f}^D$ compare to a theoretical predictor $\hat{f}$ obtained by ERM on the unseen streamed data ?

Our work is, to our knowledge, the first to address these critical questions. The last point is particularly crucial, since it determines how temporal sampling affects prediction performance which is a central question both from a theoretical perspective and for practitioners.

**Contribution.** Our contribution is threefold. First, we use a Lipschitz-based argument to obtain a sampling-dependant generalization bound for NCDE. In a second time, we build on a CDE-based well-specified model to obtain a precise decomposition of the approximation and sampling bias. With these techniques at hand, we are able to precisely assess how approximation and generalization are affected by irregular sampling. Our theoretical results are illustrated by experiments on synthetic data.

**Overview.** Section 2 covers related works. Section 3 details our setup and assumptions. In Section 4, we state our first result which is a generalization bound for NCDE. Section 5 gives an upper bound on the total risk, which is our second major result. Finally, we provide numerical illustrations in Section 6.

## 2 RELATED WORKS

**Hybridization of deep learning and differential equations.** The idea of combining differential systems and deep learning algorithms has been around for many years (Rico-Martinez et al., 1992; 1994). The recent work of Chen et al. (2018) has revived this idea by proposing a model which learns a representation of $x \in \mathbb{R}^d$ by using it to set the initial condition $z_0 = \varphi^\star_\xi(x)$ of the ordinary differential equation (ODE)

$$dz_t = \mathbf{G}_\psi(z_t)dt,$$

where $\mathbf{G}_\psi$ and $\varphi^\star_\xi$ are neural networks. The terminal value $z_1$ of this ODE is then used as input of any classical machine learning algorithm. Numeral contributions have build upon this idea and studied the connection between deep learning and differential equations in recent years (Dupont et al., 2019; Chen et al., 2019; 2020b; Finlay et al., 2020). Massaroli et al. (2020) and Kidger (2022) offer comprehensive introductions to this topic.

**Learning with irregular time series.** Numerous models have been proposed to handle time-dependant irregular data. Che et al. (2018) introduce a modified Gated Recurrent Unit (GRU) with learnt exponential decay of the hidden state between sampling times. Other models hybridizing classical deep learning architectures and ODE include GRU-ODE (De Brouwer et al., 2019) and RNN-ODE (Rubanova et al., 2019).

Neural Controlled Differential Equations have been introduced through two distinct frameworks, namely Neural Rough Differential Equations (Morrill et al., 2021b) and NCDEs (Kidger et al., 2020). In this article, we focus on the latter which extends neural ODEs to sequential data by first interpolating a time series $\mathbf{x}^D$ with cubic splines. The first value of the time series and the interpolated path $\tilde{\mathbf{x}}$ are then used as initial condition and driving signal of the CDE

$$dz_t = \mathbf{G}_\psi(z_t)d\tilde{\mathbf{x}}_t$$

or equivalently

$$z_t = \varphi_\xi^\star(x_0) + \int_0^t \mathbf{G}_\psi(z_s)d\tilde{\mathbf{x}}_s.$$

We refer to Appendix A.1 for a formal definition. As for neural ODEs, the terminal value of this NCDE is then used for classification or regression. Other methods for learning from irregular time series include Gaussian Processes (Li & Marlin, 2016) and the signature transform (Chevyrev & Kormilitzin, 2016; Fermanian, 2021; Bleistein et al., 2023).

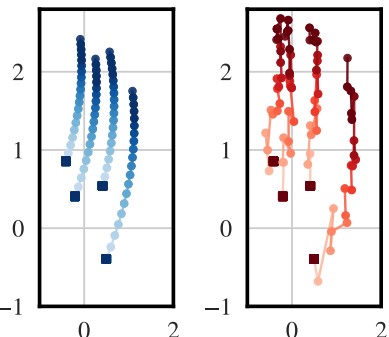

Figure 1: 2D latent state of a CDE driven by the smooth path $x_t = (t, t)$, i.e. a regular ResNet, on the **left** and a CDE driven by a Brownian motion with drift $x_t = (t + W_t^{(1)}, t + W_t^{(2)})$ on the **right**. The square markers indicate the initial values of the latent states. The color gradient indicates evolution through time.

**Generalization Bounds.** The generalization capacities of recurrent networks have been studied since the end of the 1990s, mainly through bounding their VC dimension (Dasgupta & Sontag, 1995; Koiran & Sontag, 1998). Bartlett et al. (2017) sparked a line of research connecting the Lipschitz constant of neural networks to their generalization capacities. Chen et al. (2020a) leverage this technique to derive generalization bounds for RNN, LSTM and GRU. Fermanian et al. (2021) also obtain generalization bounds for RNN and a particular class of NCDE. However, they crucially impose restrictive regularity conditions which then allow for linearization of the NCDE in the signature space. Generalization bounds are then derived using kernel learning theory. Hanson & Raginsky (2022) consider an encoder-decoder setup involving excess risk bounds on neural ODEs. Closest to our work is the recent work of Marion (2023), which leverages similar tools to study the generalization capacities of a particular class of neural ODE and deep ResNets. Our work is the first one analysing NCDEs. Recent related work is summarized in Table 1, and a detailed comparison between ResNets and NCDEs is given in Appendix F.4 - see also Figure 1 for an illustration. We refer to Appendix F.2 and F.3 for a detailed comparison with Golowich et al. (2018) and Chen et al. (2020a).

| Article | Data | Model | Proof Technique | Approximation Error |
|---|---|---|---|---|
| Bartlett et al. (2017) | Static | $H(x) = W_L \circ \sigma \circ W_{L-1} \circ \cdots \circ W_1(x)$ (MLP) | Lipschitz continuity w.r.t. to $(W_i)_{i=1}^L$ | ✗ |
| Chen et al. (2020a) | Sequential | $H_{t+1} = \sigma(AH_t + BX_t + C)$ (RNN) | Bartlett et al. (2017) | ✗ |
| Fermanian et al. (2021) | Sequential | $H_{t+1} = H_t + \frac{1}{L}\sigma(AH_t + BX_t + C)$ (Residual RNN) | Kernel-based generalization bounds | ✗ |
| Hanson & Raginsky (2022) | Static | $x \approx e^{a_m(x)f_m} \circ \cdots \circ e^{a_1(x)f_1}\xi$ (ODE-Autoencoder) | Control theory bound | NA |
| Marion (2023) | Static | $dH_t = W_t\sigma(H_t)dt$ (Neural ODE) $H_{t+1} = H_t + \frac{1}{L}W_{t+1}\sigma(H_t)$ (ResNet) | Bartlett et al. (2017) | ✗ |
| **This article** | Sequential | $dH_t = \mathbf{G}_\theta(H_t)dX_t$ (Neural CDE) $H_{t+1} = H_t + \mathbf{G}_\theta(H_t)\Delta X_{t+1}$ (Controlled ResNet) | Bartlett et al. (2017) + Continuity of the flow | ✓ |

Table 1: Summary of recent related works on generalization bounds. We do not include prior results leveraging VC dimensions. We refer to the cited work for precise definitions.

**Approximation Bounds.** The approximation capacities of neural ODEs have notably been studied by Zhang et al. (2020) and Ruiz-Balet & Zuazua (2023). In their seminal work on NCDE, Kidger et al. (2020) show that their are dense in the space of continuous functions acting on time series. In contrast to this work, we provide a precise decomposition of the approximation error when the outcome is generated by an unknown controlled dynamical system.

## 3 LEARNING WITH CONTROLLED DIFFERENTIAL EQUATIONS

### 3.1 NEURAL CONTROLLED DIFFERENTIAL EQUATIONS

We first define Neural Controlled Differential Equations driven by a very general class of paths.

**Definition 3.1.** Let $x := (x_t)_{t \in [0,1]} \in \mathbb{R}^d$ be a continuous path and let $\mathbf{G}_\psi : \mathbb{R}^p \to \mathbb{R}^{p \times d}$ be a neural vector field parameterized by $\psi$. Consider the solution $z := (z_t)_{t \in [0,1]} \in \mathbb{R}^p$ of the controlled differential equation

$$dz_t = \mathbf{G}_\psi(z_t)dx_t,$$

with initial condition $z_0 = \varphi_\xi(x_0)$, where $\varphi_\xi : \mathbb{R}^d \to \mathbb{R}^p$ is a neural network parameterized by $\xi$. We call $z$ the *latent space trajectory* and

$$\Phi^\top z_1 =: f_\theta(x) \in \mathbb{R},$$

for $\Phi \in \mathbb{R}^p$, the *prediction* of the NCDE with parameters $\theta = (\Phi, \psi, \xi)$.

The Picard-Lindelöf Theorem, recalled in Appendix A.2), ensures that the solution of this NCDE is well defined if the path $(x_t)_{t \in [0,1]}$ is of bounded variation. We make the following hypothesis on $x$ in line with this theorem.

**Assumption 1.** The path $(x_t)_{t \in [0,1]}$ is $L_x$-Lipschitz continuous, that is $\|x_t - x_s\| \le L_x|t - s|$ for all $s, t \in [0, 1]$.

Assumption 1 indeed implies that $x$ has finite bounded variation, or put otherwise for all $t \in [0, 1]$,

$$\|x\|_{\text{1-var},[0,t]} := \sup_D \sum \left\| x_{t_{i+1}} - x_{t_i} \right\| \le L_x t \le L_x,$$

where the supremum is taken on all finite discretizations $D = \{0 = t_0 < t_1 < \cdots < t_N = t\}$ of $[0, t]$, for $N \in \mathbb{N}^*$. Turning to our data $\{(\mathbf{x}^{D,i}), y_i\}$, we make two assumptions.

**Assumption 2.** For every individual $i = 1, \ldots, n$, the time series $\mathbf{x}^{D,i}$ is a discretization of a path $(x_t^i)_{t \in [0,1]}$ satisfying Assumption 1, i.e. $\mathbf{x}_{t_k}^{D,i} = x_{t_k}^i$ for all $k = 0, \ldots, K$.

**Assumption 3.** There exists a constant $B_x > 0$ such that for every individual $i = 1, \ldots, n$, $\|x_0^i\| \le B_x$.

**Neural vector fields.** The vector field $\mathbf{G}_\psi$ can be any common neural network, since most architectures are Lipschitz continuous with respect to their input (Virmaux & Scaman, 2018). This ensures that the solution of the NCDE is well defined. We restrict our attention to deep feed-forward neural vector fields with $q$ hidden layers of the form

$$\mathbf{G}_\psi(z) = \sigma\Big(\mathbf{A}_q \sigma\Big(\mathbf{A}_{q-1} \sigma\Big(\ldots \sigma\Big(\mathbf{A}_1 z + \mathbf{b}_1\Big)\Big) + \mathbf{b}_{q-1}\Big) + \mathbf{b}_q\Big), \tag{1}$$

with $L_\sigma$-Lipschitz activation function $\sigma : \mathbb{R} \to \mathbb{R}$ acting entry-wise that verifies $\sigma(0) = 0$. Our framework can be extended to architectures with different activation functions: in this case, $L_\sigma$ will be defined as the maximal Lipschitz constant. Also note that the last activation function can be taken to be the identity. For every $h = 1, \ldots, q$, $\mathbf{A}_h : \mathbb{R}^{p_{h-1}} \to \mathbb{R}^{p_h}$ is a linear operator and $\mathbf{b} \in \mathbb{R}^{p_h}$ a bias term. Since the neural vector field $\mathbf{G}_\psi$ maps $\mathbb{R}^p$ to $\mathbb{R}^{p \times d}$, the dimensions of the first and last hidden layer must verify $p_0 = p$ and $p_{q-1} = dp$. To alleviate notations and without loss of generality, we fix the size of hidden layers (but the last one) to be uniformly equal to $p$. Put otherwise, for every $h = 1, \ldots, q - 1$, $\mathbf{A}_h \in \mathbb{R}^{p \times p}$ and $\mathbf{b}_h \in \mathbb{R}^p$, while $\mathbf{A}_q \in \mathbb{R}^{p \times dp}$ and $\mathbf{b}_q \in \mathbb{R}^{p \times d}$.

Following Kidger (2022), we restrict ourselves to initializations of the form

$$z_0 = \varphi_\xi^\star(x_0) = \sigma(\mathbf{U}x_0 + v), \tag{2}$$

where $\mathbf{U} \in \mathbb{R}^{p \times d}$ and $v \in \mathbb{R}^p$. We use the notation $\mathrm{NN}_{\mathbf{U},v} : u \mapsto \sigma(\mathbf{U}u + v)$ to refer to this initialization network. The activation function is assumed to be identical for the initialization layer and the neural vector field, but such an assumption can be relaxed. The learnable parameters of the model are therefore the linear readout $\Phi$, the hidden parameters $\psi = \left(\mathbf{A}_q, \ldots, \mathbf{A}_1, \mathbf{b}_q, \ldots, \mathbf{b}_1\right)$, and the initialization weights $\xi = (\mathbf{U}, v)$. We write $\theta = \left(\Phi, \psi, \xi\right)$ for the collection of these parameters.

**Learning with time series.** To apply NCDEs to time series $\mathbf{x}^D$, one needs to embed $\mathbf{x}^D$ into the space of paths of bounded variation. This can be done through any reasonable embedding mapping

$$\rho : \mathbf{x}^D \in \left(\mathbb{R}^d\right)^{K+1} \mapsto (\tilde{\mathbf{x}}_t)_{t \in [0,1]}$$

such as splines, polynomials or linear interpolation. In this work, we focus on the fill-forward embedding, which simply defines the value of $\tilde{\mathbf{x}}$ between two consecutive points of $D$ as the last observed value of $\mathbf{x}^D$.

**Definition 3.2.** The fill-forward embedding of a time series $\mathbf{x}^D = (\mathbf{x}_{t_0}, \ldots, \mathbf{x}_{t_K})$ sampled on $D = \{0 = t_0 < \cdots < t_K = 1\}$ is the piecewise constant path $(\tilde{\mathbf{x}}_t)_{t \in [0,1]}$ defined for every $t \in [t_k, t_{k+1}[$ as $\tilde{\mathbf{x}}_t = \mathbf{x}_{t_k}^D$ for $k \in \{0, \ldots, K-1\}$, and $\tilde{\mathbf{x}}_1 = \mathbf{x}_{t_K} = x_1$.

Using this embedding in a NCDE with parameters $\theta = (\Phi, \psi, \xi)$ recovers a piecewise constant latent space trajectory $z^D := (z_t^D)_{t \in [0,1]}$ recursively defined for every $t \in [t_k, t_{k+1}[$ by

$$z_t^D = z_{t_{k-1}}^D + \mathbf{G}_\psi(z_{t_{k-1}})(\mathbf{x}_{t_k} - \mathbf{x}_{t_{k-1}}) \tag{3}$$

for $k = 1, \ldots, K-1$ and initialized as $z_0^D := z_0 = \sigma(\mathbf{U}x_0 + v)$.

Formally, the prediction $\Phi^\top z_1^D$ is equal to $\Phi^\top z_1^D = f_\theta \circ \rho_{\text{FF}}\left(\mathbf{x}^D\right)$ where $\rho_{\text{FF}}$ is the fill-forward operator. To lighten notations, we simply write $\Phi^\top z_1^D = f_\theta(\mathbf{x}^D)$. This recursive architecture has been studied with random neural vector fields by Cirone et al. (2023) under the name of *homogenous controlled ResNet* because of its resemblance with the popular weight-tied ResNet (He et al., 2015). We highlight that the latent continuous state $z$ is more informative about the outcome $y$ than $z^D$: it embeds the full trajectory of $x$ while $z^D$ embeds a version of $x$ degraded through sampling.

**Restrictions on the parameter space.** In order to obtain generalization bounds, we must furthermore restrict the size of parameter space by requiring that the parameters $\theta$ lie in a bounded set $\Theta$. This means that there exist $B_\Phi, B_\mathbf{A}, B_\mathbf{b}, B_\mathbf{U}, B_v$ such that

$$\|\mathbf{A}_h\| \leq B_\mathbf{A}, \|\mathbf{b}_h\| \leq B_\mathbf{b}, \|\mathbf{U}\| \leq B_\mathbf{U}, \|v\| \leq B_v, \|\Phi\| \leq B_\Phi \tag{4}$$

for all $h = 1, \ldots, q$ and for all NCDEs considered, where $\|\cdot\|$ is the Frobenius norm. Such restrictions are classical for deriving generalization bounds (Bartlett et al., 2017; Bach, 2021; Fermanian et al., 2021). We call

$$\mathcal{F}_\Theta = \left\{ f_\theta : (x_t)_{t \in [0,1]} \mapsto f_\theta(x) \in \mathbb{R} \;\; \text{s.t.} \;\; \theta \in \Theta \right\}$$

this class of predictors. We also write

$$\mathcal{F}_\Theta^D = \left\{ f_\theta : \mathbf{x}^D \in \mathbb{R}^{d \times |D|} \mapsto f_\theta(\mathbf{x}^D) \in \mathbb{R} \;\; \text{s.t.} \;\; \theta \in \Theta \right\}$$

for the class of homogenous controlled ResNets. We now state an important lemma, which ensures that the outputs of NCDEs remain bounded. It is a direct consequence of Gronwall's Lemma, recalled in Lemma A.4. We let $|D| := \max_{j=1,\ldots,K-1} |t_{j+1} - t_j|$ be the greatest gap between two sampling times.

**Lemma 3.3.** The value $f_\theta(x)$ is uniformly upper bounded by

$$M_\Theta := B_\Phi L_\sigma \exp\left((B_\mathbf{A} L_\sigma)^q L_x\right)\left(B_\mathbf{U} B_x + B_v + \kappa_\Theta(\mathbf{0}) L_x\right)$$

and $f_\theta(\mathbf{x}^D)$ is upper bounded by

$$M_\Theta^D := B_\Phi L_\sigma \left(1 + \left(L_\sigma B_\mathbf{A}\right)^q L_x |D|\right)^K \left(B_\mathbf{U} B_x + B_v + \kappa_\Theta(\mathbf{0}) L_x\right),$$

where $\kappa_\Theta(\mathbf{0}) = L_\sigma B_\mathbf{b} \frac{\left(L_\sigma B_\mathbf{A}\right)^q - 1}{L_\sigma B_\mathbf{A} - 1}$ is an upper bound of $\|\mathbf{G}_\psi(\mathbf{0})\|_{\text{op}} =: \max_{\|u\|=1} \|\mathbf{G}_\psi(0)u\|$.

**Remark 3.4.** Remark that the constant $M_\Theta^D$ grows exponentially with $q$ if $B_\mathbf{A} L_\sigma > 1$. In practice, NCDEs are however used with shallow neural vector fields, i.e. $q$ is typically chosen to be less than 3 (Kidger et al., 2020). Additionally, if one wants to use very deep neural vector fields, this result suggest controlling their Frobenius norm by setting it equal to 1 to avoid exponential blow-up of the network's Lipschitz constant - see for instance Li et al. (2019).

**Remark 3.5.** It is natural to wonder whether $M_\Theta^D \to M_\Theta$ as $|D| \to 0$. This is indeed true for well-behaved samplings, i.e. when adding more sampling points decreases the mesh size $|D|$ at sufficient speed. Indeed, if $K^{-1} = \mathcal{O}(|D|)$, one can use the fact that $(1 + \alpha x^{-1})^x \to \exp(\alpha)$ as $x \to 0$ to see that $M_\Theta^D \to M_\Theta$.

### 3.2 THE LEARNING PROBLEM

We now detail our learning setup. We consider an i.i.d. sample $\{(y^i, \mathbf{x}^{D,i})\}_{i=1}^n \sim y, x$ with given discretization $D$. For a given predictor $f_\theta \in \mathcal{F}_\Theta$, define

$$\mathcal{R}_n^D(f_\theta) = \frac{1}{n}\sum_{i=1}^n \ell(y^i, f_\theta(\mathbf{x}^{D,i})) \quad \text{and} \quad \mathcal{R}^D(f_\theta) = \mathbb{E}_{x,y}\Big[\ell(y, f_\theta(\mathbf{x}^D))\Big]$$

as the empirical risk and the expected risk on the discretized data. Similarly, define

$$\mathcal{R}_n(f_\theta) = \frac{1}{n}\sum_{i=1}^n \ell(y^i, f_\theta(x^i)) \quad \text{and} \quad \mathcal{R}(f_\theta) = \mathbb{E}_{x,y}\Big[\ell(y, f_\theta(x))\Big]$$

as the empirical risk and expected risk on the continuous data. We stress that $\mathcal{R}_n(f_\theta)$ cannot be optimized, since we do not have access to the continuous data. Let $\hat{f}^D \in \arg\min_{\theta \in \Theta} \mathcal{R}_n^D(f_\theta)$ be an

optimal predictor obtained by empirical risk minimization on the discretized data. In order to obtain generalization bounds, the following assumptions on the loss and the outcome are necessary (Mohri et al., 2018).

**Assumption 4.** The outcome $y \in \mathbb{R}$ is bounded a.s.

**Assumption 5.** The loss $\ell : \mathbb{R} \times \mathbb{R} \to \mathbb{R}_+$ is Lipschitz with respect to its second variable, that is there exists $L_\ell$ such that for all $u, u' \in \mathcal{Y}$ and $y \in \mathcal{Y}$, $|\ell(y, u) - \ell(y, u')| \le L_\ell |u - u'|$.

This hypothesis is satisfied for most classical losses, such as the the mean squared error, as long as the outcome and the predictions are bounded. This is true by Assumption 4 and Lemma 3.3. The loss function is thus bounded since it is continuous on a compact set, and we let $M_\ell$ be a bound on the loss function.

## 4 A GENERALIZATION BOUND FOR NCDE

We now state our main theorem.

**Theorem 4.1.** *With probability at least $1 - \delta$, the generalization error $\mathcal{R}^D(\hat{f}^D) - \mathcal{R}_n(\hat{f}^D)$ is upper bounded by*

$$\frac{24 M_\Theta^D L_\ell}{\sqrt{n}}\sqrt{2pU_1^D + (q-1)p(p+1)U_2^D + dp(2+p)U_3^D} + M_\ell\sqrt{\frac{\log 1/\delta}{2n}},$$

*with $U_1^D := \log(\sqrt{n}C_q K_1^D)$, $U_2^D := \log(\sqrt{np}C_q K_2^D)$, $U_3^D = \log(\sqrt{ndp}C_q K_2^D)$, and $C_q := (8q + 12)$. $K_1^D$ and $K_2^D$ are two discretization and depth dependant constants equal to*

$$K_1^D := \max\{B_\Phi M_\Theta^D, B_v C_v\} \text{ and } K_2^D := \max\{B_\mathbf{b}C_\mathbf{b}^D, B_\mathbf{A}C_\mathbf{A}^D, B_\mathbf{U}C_\mathbf{U}\},$$

*where $C_\mathbf{A}^D, C_\mathbf{b}^D, C_v, C_\mathbf{U}$ are Lipschitz constants given in Appendix D.7.*

This result is similar to results obtained for instance by Bartlett et al. (2017), Chen et al. (2020a) and Marion (2023) for different models. Indeed, the generalization error is upper bounded by a noise induced term and a complexity term which grows with the square root of the model's complexity measured by the number of hidden layers $q$ and the dimension $p$ of the hidden state. This theorem is obtained by upper bounding the covering number of $\mathcal{F}_\Theta^D$ and the Rademacher complexity of NCDEs.

**Proposition 4.2.** *When using discretized inputs, the covering number $\mathcal{N}(\mathcal{F}_\Theta, \eta; D)$ of $\mathcal{F}_\Theta^D$ is bounded from above by*

$$\left(1 + \frac{C_q K_1^D}{2\eta}\sqrt{p}\right)^{(q+1)p(1+p)} \left(1 + \frac{C_q K_2^D}{2\eta}(\sqrt{p} + \sqrt{d})\right)^{2dp} \left(1 + \frac{C_q K_1^D}{2\eta}\sqrt{dp}\right)^{dp^2}.$$

*Furthermore, the empirical Rademacher complexity $\mathrm{Rad}(\mathcal{F}_\Theta^D)$ associated to an i.i.d. sample of size $n$ discretized on $D$ is upper bounded by*

$$\frac{4}{n} + \frac{24 M_\Theta^D}{\sqrt{n}}\sqrt{(q+1)p(p+1)U_1^D + 2dpU_2^D + dp^2 U_3^D}.$$

*When using continuous inputs, the constants appearing in the upper bound of $\mathrm{Rad}(\mathcal{F}_\Theta)$ are discretization independent.*

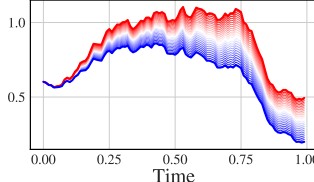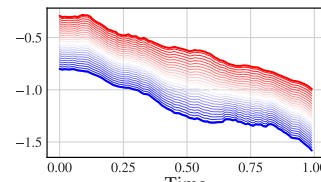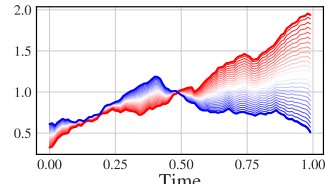

Figure 2: **Left:** in bold, we plot the solutions of two shallow ($q = 1$) NCDEs who only differ in their vector field $\mathbf{A}^1$ and $\mathbf{A}^2$. We then plot the solutions of the NCDEs with interpolated vector field $\delta\mathbf{A}^1 + (1 - \delta)\mathbf{A}^2$ for $\delta \in [0, 1]$ in red-blue gradient. **Center:** we consider a given NCDE and interpolate linearly between two initial conditions $x_0^1$ and $x_0^2$. **Right:** we consider a given NCDE and drive it with the linear interpolation of two paths $(x_t^1)_t$ and $(x_t^2)_t$. In all three cases, the solutions evolve continuously as we interpolate between the models.

This result is obtained by showing that NCDE are Lipschitz with respect to their parameters, such that covering each parameter class yields a covering of the whole class of predictors. In combination with arguments borrowed from Chen et al. (2020a) and Bartlett et al. (2017), one may then upper bound the empirical Rademacher complexity by making use of the Dudley Integral.

## 5 A Bound on the Total Risk

### 5.1 A CDE-Based Well-Specified Model

In order to bound the approximation bias, we introduce a model that links the underlying path $x$ to the outcome $y$.

**Assumption 6.** There exists a vector field $\mathbf{G}^* : \mathbb{R}^p \to \mathbb{R}^{p \times d}$, a function $\varphi^\star : \mathbb{R}^d \to \mathbb{R}^p$ and a vector $\Phi_\star \in \mathbb{R}^p$ that govern a latent state $(z_t^\star)_{t \in [0,1]}$ through the CDE

$$dz_t^\star = \mathbf{G}^\star(z_t^\star)dx_t \tag{5}$$

with initial value $z_0^\star = \varphi^\star(x_0)$ such that the outcome $y$ is given by

$$y = \Phi_\star^\top z_1^\star + \varepsilon,$$

where $\varepsilon$ is a noise term bounded by $M_\varepsilon > 0$.

**Assumption 7.** The parameters of the well-specified model satisfy $\|\Phi_\star\| \leq B_\Phi$, $\varphi^\star \in C(\mathbb{R}^d, \mathbb{R}^p)$ and $\mathbf{G}^\star \in \mathrm{Lip}(\mathbb{R}^p, \mathbb{R}^{p \times d})$ with Lipschitz constant $L_{\mathbf{G}^\star}$.

**Remark 5.1.** This is a very general model, which encompasses ODE-based models used for instance in biology or physics. Indeed, by setting $x_t = t$ for all $t$, one recovers a generic ODE. It also allows for the modelization of complex dynamical systems whose dynamics depend on external time-dependant data. For instance, Simeoni et al. (2004) model tumor growth kinetics as a CDE driven by a time series recording the intake of anticancer agents, which perturbs the dynamics of the growth of the tumor.

We call the map $\mathbf{f}^\star : x \mapsto \Phi_\star^\top z_1^\star$ the true predictor. Uniqueness and existence to equation 5 is given by the Picard-Lindelöf theorem stated in Appendix A.2. We have the following important lemma, which ensures that outcomes of the well-specified model are bounded. Its is analogous to Lemma 3.3.

**Lemma 5.2.** *Let $y$ be generated from the CDE (5) from a $L_x$-Lipschitz path $x$. Let $B_\varphi^\star := \max_{\|u\| \leq B_x} \|\varphi^\star(u)\|$. Then Assumption 4 holds, i.e. $y$ is bounded, and*

$$|y| \leq B_\Phi \left( B_\varphi^\star + \|\mathbf{G}^\star(0)\|_{\mathrm{op}} L_x \right) \exp\left( L_{\mathbf{G}^\star} L_x \right) + M_\varepsilon.$$

### 5.2 Bounding the Total Risk

We first decompose the difference between the expected risk of $\hat{f}^D$ learnt from the sample $S$ and the expected risk of $\mathbf{f}^*$.

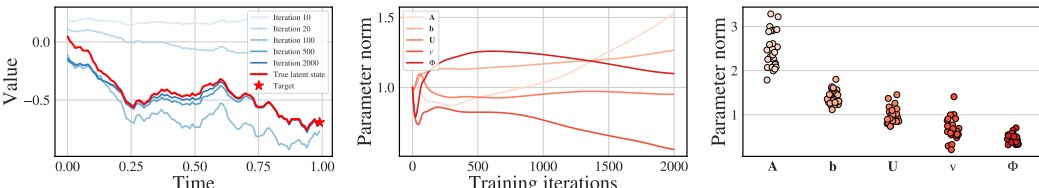

Figure 3: **Left**: Evolution of the prediction $(\Phi^\top z_t)_t$ of a NCDE with shallow neural vector field at different moments of the training process ; the true process $(\Phi_\star^\top z_t^\star)_t$ is shown in red. **Center:** Evolution of the parameter's norms, normalized by their value at initialization, during training. **Right:** Parameters' norm at the end of training, *without* normalization, over 25 training instances of the same model. Training is performed with Adam (Kingma & Ba, 2014).

**Lemma 5.3.** *For all $f_\theta \in \mathcal{F}_\theta$, the total risk $\mathcal{R}^D(\hat{f}^D) - \mathcal{R}(\mathbf{f}^*)$ is almost surely bounded from above by*

$$\sup_{g \in \mathcal{F}_\Theta^D} \left[ \mathcal{R}^D(g) - \mathcal{R}_n^D(g) \right] + \sup_{g \in \mathcal{F}_\Theta} \left[ \mathcal{R}(g) - \mathcal{R}_n(g) \right] + \underbrace{\mathcal{R}_n^D(f_\theta) - \mathcal{R}_n(f_\theta)}_{\text{Discretization bias}} + \underbrace{\mathcal{R}(f_\theta) - \mathcal{R}(\mathbf{f}^*)}_{\text{Approximation bias}}.$$

Turning to the total risk, we get the following result by bounding every term of Lemma 5.3.

**Theorem 5.4.** *The total risk $\mathcal{R}^D(\hat{f}^D) - \mathcal{R}(\mathbf{f}^*)$ is bounded from above by*

$$4 \max\{\text{Rad}(\mathcal{F}_\Theta), \text{Rad}(\mathcal{F}_\Theta^D)\} \qquad\qquad\qquad\qquad\text{(Worst-case bound)}$$

$$+ L_\ell B_\Phi L_x \left[ 1 + (L_\sigma B_\mathbf{A})^q \left( (L_\sigma B_\mathbf{A})^q \frac{\min\{M_\Theta^D, M_\Theta\}}{B_\Phi} + \kappa_\Theta(\mathbf{0}) \right) L_x \right] |D| \quad \text{(Discretization Bias)}$$

$$+ L_\ell B_\Phi \exp(L_{\mathbf{G}^\star} L_x) \left[ L_x \min_\psi \max_{u \in \Omega} \|\mathbf{G}_\psi(u) - \mathbf{G}^\star(u)\|_{\text{op}} + \min_{\mathbf{U},v} \max_{\|u\| \le B_x} \|\varphi^\star(u) - \text{NN}_{\mathbf{U},v}(u)\| \right].$$
$$\text{(Approximation Bias)}$$

Let us comment the three terms of the bound of Theorem 5.4. The first term is a common worst case bound when deriving generalization bounds and is bounded using Proposition 4.2. The second term corresponds to the discretization bias, and is directly proportional to $|D|$. It therefore vanishes at the same speed as $|D|$. For instance, if we consider the sequence $D_K$ of equidistant discretizations of $[0, 1]$ with $K$ points, the discretization bias vanishes at linear speed. Finally, the approximation bias writes as the sum of approximation errors on the vector fields and the initial condition, rather than a general approximation error on the true predictor $\mathbf{f}^\star$. The diameter of $\Omega$ is upper bounded in Proposition C.2.

**Remark 5.5.** This decomposition of the approximation bias allows to leverage approximation results for feed forward neural networks to obtain precise bounds (see for instance Berner et al., 2021; Shen et al., 2021). This is left for future work.

**Outline of the proof.** The proof schematically works in two times. We first bound the two sources of bias by leveraging the continuity of the flow of a CDE. Informally, this theorem states that the difference $\Delta$(terminal value) between the terminal value of two CDEs decomposes as

$$\Delta(\text{vector fields}) + \Delta(\text{initial conditions}) + \Delta(\text{driving paths}).$$

This neat decomposition allows us to bound the discretization bias since it depends on the terminal value of two CDEs whose driving paths differ. This property is illustrated in Figure 2. Combining these results with Theorem 4.1 gives Theorem 5.4.

## 6 NUMERICAL ILLUSTRATIONS

**Training Dynamics.** Our proofs rely on the restriction that all considered predictors, including $\hat{f}^D$ obtained by ERM, have parameters lying in a bounded set $\Theta$. To check whether this hypothesis

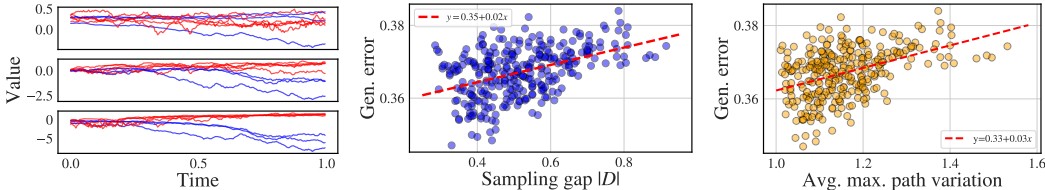

Figure 4: **Left:** latent state $(\Phi^\top z_t)_t$ at initialization (top) and after 10 (middle) resp. 100 (bottom) training steps as the NCDE learns to separate rough (in red) from smooth (in blue) paths on a fine grid. The figure shows examples from the test set. **Center**: Generalization error $|n^{-1}\sum \ell(\mathbf{x}^{D,i}, y^i) - \mathbb{E}_{\mathbf{x}^D,y}\,\ell(\mathbf{x}^D, y)|$ vs. sampling gap. **Right**: Generalization error vs. average maximal path variation. We train 300 NCDEs, with $\Phi$ and the initialization layer left untrained to isolate the effect of the discretization, with time series downsampled on $K = 5$ points randomly chosen in $[0,1]$ for each run. Training is done with Adam with default parameters (Kingma & Ba, 2014).

is reasonable, we examine the evolution of the parameters' norms during training. Figure 3 displays the evolution of the latent state of a shallow model ($q = 1$) through training, along with the evolution of the normalized parameter norm, in a teacher-student i.e. well-specified setup. The trained NCDE achieves perfect interpolation of the outcome but also, more surprisingly, almost perfect interpolation of the unobserved latent state. We also observe little dispersion of the parameters' norm over multiple training instances.

**Effect of the Discretization.** Our bounds highlight the effect of the discretization gap $|D|$ on the generalization capacities of NCDEs. In our second experiment, presented in Figure 4, we consider a classification task in which a NCDE is trained to distinguish between rough ($H = 0.4$) and smoother ($H = 0.6$) discretized sample paths of a fractional Brownian Motion (fBM) with Hurst exponent $H$. We randomly downsample the paths, train the model and compute the generalization error in terms of binary cross entropy loss. We see a clear correlation between the generalization error and the sampling gap $|D|$. Since our bound involving $|D|$ relies on the inequality

$$\max_{k=1,\ldots,K} \left\| \mathbf{x}_{t_{k+1}}^{D,i} - \mathbf{x}_{t_k}^{D,i} \right\| \leq \max_{k=1,\ldots,K} L_x |t_{k+1} - t_k| \leq L_x |D|,$$

we also check for correlation with the average maximal path variation

$$n^{-1} \sum_{i=1}^{n} \max_{k=0,\ldots,K} \left\| \mathbf{x}_{t_{k+1}}^{D,i} - \mathbf{x}_{t_k}^{D,i} \right\|.$$

As predicted, this quantity also correlates with the generalization error. This is consistent with the findings of Marion (2023), who observe that the generalization error of ResNets scales with the Lipschitz constant of the weights (see Appendix F.4 for a detailed discussion). All experimental details are given in Appendix E.

## 7    CONCLUSION

We have answered three open theoretical questions on NCDE. First, we have provided a generalization bound for this class of models. In a second time, we have shown that under the hypothesis of CDE-based well-specified model, the approximation bias of NCDE decomposes into a sum of tractable biases. Most importantly, all our results are discretization-dependent and allow to precisely quantify the theoretical impact of the time series sampling on the model's performance. Practical implications for the design of NCDEs and control theory are discussed in Appendix F.1.

**Acknowledgements.** We thank Pierre Marion and Adeline Fermanian for proofreading and providing insightful comments and remarks, and Cristopher Salvi for discussions on Controlled ResNets. LB conducted parts of this research while visiting MBZUAI's Machine Learning Department (UAE).

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

The appendix is structured as follows. Appendix A gives preliminary results on CDEs and covering numbers. Appendix B.3 collects the proofs of the generalization bound. Appendix C details the proof of the bounds of the approximation and discretization biases and the bound on the total risk. Proofs of Appendix A are given in Appendix D for the sake of clarity. Appendix E gives all experimental details.

## A  MATHEMATICAL BACKGROUND

### A.1  RIEMANN-STIELTJES INTEGRAL

**Lemma A.1.** *Let $f : [0,1] \to \mathbb{R}^{p \times d}$ be a continuous function, and $g : [0,1] \to \mathbb{R}^d$ be of finite total variation. Then Riemann-Stieljes integral*

$$\int_0^t f_s dg_s \in \mathbb{R}^p,$$

*where the product is understood in the sense of matrix-vector multiplication, is well-defined and finite for every $t \in [0,1]$.*

We refer to Friz & Victoir (2010) for a thorough introduction to the Riemann-Stieltjes integral.

### A.2  PICARD-LINDELÖF THEOREM

**Theorem A.2.** *Let $x$ be a continuous path of bounded variation, and assume that $\mathbf{G} : \mathbb{R}^p \to \mathbb{R}^{p \times d}$ is Lipschitz continuous. Then the CDE*

$$dz_t = \mathbf{G}(z_t)dx_t$$

*with initial condition $z_0 = \varphi^\star(x_0)$ has a unique solution.*

A full proof can be found in Fermanian et al. (2021). Remark that since in our setting, NCDEs are Lipschitz since the neural vector fields we consider are Lipschitz (Virmaux & Scaman, 2018). This ensures that the solutions to NCDEs are well defined.

We also need the following variation, which ensures that the CDE driven by $\tilde{\mathbf{x}}$ is well-defined.

**Lemma A.3.** *Let $x$ be a piecewise constant and right-continuous path taking finite values in $\mathbb{R}^d$, with a finite number of discontinuities at $0 < t_1, \ldots, t_K = 1$, i.e.*

$$\lim_{t \to t_i^+} x_t = x_{t_i}$$

*and*

$$x_t = x_{t_i}$$

*for all $t \in [t_i, t_{i+1}[$, for all $i = 1, \ldots, K-1$. Assume that $\mathbf{G} : \mathbb{R}^p \to \mathbb{R}^{p \times d}$ is Lipschitz continuous. Then the CDE*

$$dz_t = \mathbf{G}(z_t)dx_t$$

*with initial condition $z_0 = \varphi^\star(x_0)$ has a unique solution.*

This result can be obtained by first remarking that since $x$ is piecewise constant, the solution to this CDE will also be piecewise constant, with discontinues at $0 < t_1 < \cdots < t_K$: indeed, the variations of $x$ between two points of discontinuity being null, the variations of the solution will also be null between these two points. The solution can then be recursively obtained by seeing that for all $t \in [t_i, t_{i+1}[$

$$z_t = z_{t_i} = z_{t_{i-1}} + \mathbf{G}(z_{t_{i-1}})(\mathbf{x}_{t_i} - \mathbf{x}_{t_{i-1}})$$

with $z_t = \varphi^\star(z_0)$ for $t \in [t_0, t_1[$, where $t_0 = 0$, and $z_{t_K} = y_1 = y_{t_{K-1}} + \mathbf{G}(y_{t_{K-1}})(\mathbf{x}_{t_K} - \mathbf{x}_{t_{K-1}})$.

## A.3 Gronwall's Lemmas

**Lemma A.4** (Gronwall's Lemma for CDEs). *Let $x : [0,1] \to \mathbb{R}^d$ be a continuous path of bounded variations, and $\phi : [0,1] \to \mathbb{R}^d$ be a bounded measurable function. If*

$$\phi(t) \le K_t + L_x \int_0^t \phi(s) \, \|dx_s\|$$

*for all $t \in [0,1]$, where $K_t$ is a time dependant constant, then*

$$\phi(t) \le K_t \exp\left(L_x \, \|x\|_{1\text{-var},[0,t]}\right)$$

*for all $t \ge 0$.*

See Friz & Victoir (2010) for a proof. Remark that this Lemma does not require $\phi(\cdot)$ to be continuous. We also state a variant of the discrete Gronwall Lemma, which will allow us to obtain the bound for discrete inputs $\mathbf{x}^D$.

**Lemma A.5** (Gronwall's Lemma for sequences). *Let $(y_k)_{k\ge 0}, (b_k)_{k\ge 0}$ and $(f_k)_{k\ge 0}$ be positive sequences of real numbers such that*

$$y_n \le f_n + \sum_{l=0}^{n-1} b_l y_l$$

*for all $n \ge 0$. Then*

$$y_n \le f_n + \sum_{l=0}^{n-1} f_l b_l \prod_{j=l+1}^{n-1} (1 + b_j)$$

*for all $n \ge 0$.*

A proof can be found in Holte (2009) and Clark (1987). We finally need a variant of Gronwall's Lemma for sequences.

**Lemma A.6.** *Let $(u_k)_{k\ge 0}$ be a sequence such that for all $k \ge 1$,*

$$u_k \le a_k u_{k-1} + b_k$$

*for $(a_k)_{k\ge 1}$ and $(b_k)_{k\ge 1}$ two positive sequences. Then for all $k \ge 1$*

$$u_k \le \Big(\prod_{j=1}^{k} a_j\Big) u_0 + \sum_{j=1}^{k} b_j \Big(\prod_{i=j+1}^{k} a_i\Big). \tag{6}$$

## A.4 Dudley's Entropy Integral

We recall the following Lemma from Bartlett et al. (2017).

**Lemma A.7.** *Let $\mathcal{H}$ be a class of real-valued functions taking values in $[-M, M]$, for $M > 0$, and assume that $0 \in \mathcal{H}$. Then the empirical Rademacher complexity associated to a sample of $n$ datapoints verifies*

$$\text{Rad}(\mathcal{H}) \le \inf_{\eta > 0} \left[\frac{4\eta}{\sqrt{n}} + \frac{12}{n} \int_{\eta}^{2M\sqrt{n}} \sqrt{\log \mathcal{N}(\mathcal{H}, s)} ds\right]. \tag{7}$$

## A.5 Covering numbers

First, we recall the definition of the covering number of a class of functions.

**Definition A.8.** Let $\mathcal{H} = \{h : \mathcal{U} \subset \mathbb{R}^e \to \mathbb{R}^f\}$ be a class of functions. The covering number $\mathcal{N}(\mathcal{H}, \eta)$ of $\mathcal{H}$ is the minimal cardinality of a subset $C \subset \mathcal{H}$ such that for all $h \in \mathcal{H}$, there exists $\hat{h} \in C$ such that

$$\sup_{x \in \mathcal{U}} \left\|h(x) - \hat{h}(x)\right\| \le \eta.$$

We state a Lemma from Chen et al. (2020a) bounding the covering number of sets of linear maps.

**Lemma A.9.** *Let $\mathcal{G} = \{A \in \mathbb{R}^{f \times e} \text{ s.t. } \|A\| \leq \lambda\}$ for a given $\lambda > 0$. The covering number $\mathcal{N}(\mathcal{G}, \eta)$ is upper bounded by*

$$\left(1 + \frac{2\min(\sqrt{e}, \sqrt{f})\lambda}{\eta}\right)^{ef}.$$

We now proceed to prove our main results. Let us precisely detail the different steps of our proofs.

1. For the generalization bound, the first step consists in showing that NCDEs are Lipschitz with respect to their parameters, which we do in Theorem B.6. This leverages the continuity of the flow, since we aim at bounding the difference between two NCDEs with different parameters but identical driving paths.

2. Building on Chen et al. (2020a), we then obtain a covering number for NCDE.

3. We connect the covering number of $\mathcal{F}_\Theta$ and $\mathcal{F}_\Theta^D$ to its Rademacher complexity building on arguments made by Bartlett et al. (2017) in Proposition 4.2.

4. Once this last result is obtained, we can obtain the generalization bound by resorting to classical techniques (Mohri et al., 2018).

5. The bound of the discretization bias, stated in Proposition C.1, follows from the continuity of the flow given in Theorem B.5. Indeed, the discretization error depends on the difference between $z_1$ and $z_1^D$, which are solutions to identical CDE, but with different driving paths $x$ and embedded path $\tilde{\mathbf{x}}$.

6. The bound of the approximation bias is obtained in a similar fashion. Indeed, it directly depends on the difference between $z_1^\star$ and $z_1$. However, this time, only the initial condition and the vector fields are different, while the driving paths are identical.

7. The bound on the total risk is obtained by combining all the precedent elements and using a symmetrization argument to upper bound the two worst case bounds.

## B    PROOF OF THE GENERALIZATION BOUND

### B.1    LIPSCHITZ CONTINUITY OF NEURAL VECTOR FIELDS

We first state a series of Lemmas on the Lipschitz continuity of neural vector fields. We first prove Lipschitz continuity with respect to their inputs.

**Lemma B.1.** *Let $\mathbf{G}_\psi$ be a neural vector field of depth $q \geq 1$ as defined in Equation 1. Then for all $z, w \in \mathbb{R}^p$*

$$\|\mathbf{G}_\psi(z) - \mathbf{G}_\psi(w)\| \leq \left(L_\sigma B_\mathbf{A}\right)^q \|z - w\| \tag{8}$$

Our second Lemma shows that the operator norm of neural vector fields grows linearly with $z$.

**Lemma B.2.** *Let $\mathbf{G}_\psi : \mathbb{R}^p \to \mathbb{R}^{p \times d}$ be a neural vector field of depth $q$. For all $z \in \mathbb{R}^p$, we have*

$$\|\mathbf{G}_\psi(z)\| \leq \left(L_\sigma B_\mathbf{A}\right)^q \|z\| + L_\sigma B_\mathbf{b} \frac{\left(L_\sigma B_\mathbf{A}\right)^q - 1}{L_\sigma B_\mathbf{A} - 1} \tag{9}$$

*and in particular*

$$\|\mathbf{G}_\psi(\mathbf{0})\| \leq L_\sigma B_\mathbf{b} \frac{\left(L_\sigma B_\mathbf{A}\right)^q - 1}{L_\sigma B_\mathbf{A} - 1} = \kappa_\Theta(\mathbf{0}). \tag{10}$$

Let $\mathbf{G}_{\psi^1}$ and $\mathbf{G}_{\psi^2}$ be two deep neural vector fields, as defined in Equation 1, with learnable parameters $\psi^1 = \left(\mathbf{A}_q^1, \ldots, \mathbf{A}_1^1, \mathbf{b}_q^1, \ldots, \mathbf{b}_1^1\right)$ and $\psi^2 = \left(\mathbf{A}_q^2, \ldots, \mathbf{A}_1^2, \mathbf{b}_q^2, \ldots, \mathbf{b}_1^2\right)$. We have the following Lemma.

**Lemma B.3** (Lipschitz Continuity w.r.t. the parameters). *Let $z \in \mathbb{R}^p$ and $\mathbf{G}_{\psi^1}, \mathbf{G}_{\psi^2} : \mathbb{R}^p \to \mathbb{R}^{p \times d}$ be two neural vector fields defined as above. Then for all $z \in \mathbb{R}^p$*

$$\left\|\mathbf{G}_{\psi^1}(z) - \mathbf{G}_{\psi^2}(z)\right\|_{\mathrm{op}} \leq \sum_{j=1}^q \left[C_\mathbf{b}^j \left\|\mathbf{b}_j^1 - \mathbf{b}_j^2\right\| + C_\mathbf{A}^j(z) \left\|\mathbf{A}_j^1 - \mathbf{A}_j^2\right\|\right] \tag{11}$$

*for $(C_\mathbf{b}^j)_{j=1}^q$ and $(C_\mathbf{A}^j(z))_{j=1}^q$ two sequences of constants explicitly given in the proof.*

## B.2 AN UPPER BOUND ON THE TOTAL VARIATION OF THE SOLUTION OF A CDE

We have the following bound on the total variation of the solution to a CDE.

**Proposition B.4.** *Let $\mathbf{F} : \mathbb{R}^{p \times d} \to \mathbb{R}^d$ be a Lipschitz vector field with Lipschitz constant $L_{\mathbf{F}}$ and $x = (x_t)_{t \in [0,1]}$ be a continuous path of total variation bounded by $L_x$. Let $z = (z_t)_{t \in [0,1]}$ be the solution of the CDE*

$$dz_t = \mathbf{F}(z_t)dx_t$$

*with initial condition $z_0 \in \mathbb{R}^p$. Then for all $t \in [0, 1]$, we have that*

$$\|z\|_{\text{1-var},[0,t]} \leq C_1(\mathbf{F}) \|x\|_{\text{1-var},[0,t]}$$

*with*

$$C_1(\mathbf{F}) := \left[ L_{\mathbf{F}} \big( \|z_0\| + \|\mathbf{F}(0)\|_{\text{op}} L_x \big) \exp(L_{\mathbf{F}} L_x) + \|\mathbf{F}(0)\|_{\text{op}} \right] \exp(L_{\mathbf{F}} L_x).$$

*One also has that the total variations of $z$ and $z^D$ are both upper bounded and we have*

$$\|z\|_{\text{1-var}} \leq \left[ \left( L_\sigma B_{\mathbf{A}} \right)^q \frac{1}{B_\Phi} M_\Theta + \kappa_\Theta(\mathbf{0}) \right] \exp((L_\sigma B_{\mathbf{A}})^q L_x) L_x$$

*and*

$$\|z^D\|_{\text{1-var}} \left[ \left( L_\sigma B_{\mathbf{A}} \right)^q \frac{1}{B_\Phi} M_\Theta^D + \kappa_\Theta(\mathbf{0}) \right] \exp((L_\sigma B_{\mathbf{A}})^q L_x) L_x. \tag{12}$$

The proof is given in Appendix D.5.

## B.3 CONTINUITY OF THE FLOW OF A CDE

The following theorem is central for our analysis of the approximation bias. Its core idea is that the difference between the solution of two CDEs can be upper bounded by the sum of the differences between their vector fields, their driving paths and their initial values.

**Theorem B.5.** *Let $\mathbf{F}, \mathbf{G} : \mathbb{R}^p \to \mathbb{R}^{p \times d}$ be two Lipschitz vector fields with Lipschitz constants $L_{\mathbf{F}}, L_{\mathbf{G}}$. Let $x, r$ be either continuous or piecewise constant paths of total variations bounded by $L_x$ and $L_r$. Consider the controlled differential equations*

$$dw_t = \mathbf{F}(w_t)dx_t \quad and \quad dv_t = \mathbf{G}(v_t)dr_t$$

*with initial conditions $w_0 \in \mathbb{R}^p$ and $v_0 \in \mathbb{R}^p$ respectively. We have that*

$$\|w_t - v_t\| \leq \Bigg( \|w_0 - v_0\| + \|x_0 - r_0\|$$
$$+ \|x - r\|_{\infty,[0,t]} \left( 1 + L_{\mathbf{F}} L_r C_1(\mathbf{F}) \right) + \max_{v \in \Omega} \|\mathbf{F}(v) - \mathbf{G}(v)\|_{\text{op}} L_r \Bigg)$$
$$\times \exp(L_{\mathbf{F}} L_x).$$

*where the constant $C_1(\mathbf{F})$ is given in Proposition B.4 and the set $\Omega$ is defined as*

$$\Omega = \big\{ u \in \mathbb{R}^p \mid \|u\| \leq (\|v_0\| + \|\mathbf{G}(0)\|_{\text{op}} L_r) \exp(L_{\mathbf{G}} L_r) \big\}.$$

The proof is given in Appendix D.6. We stress that any combination of continuous and piecewise constant paths can be used.

Using this result, we obtain the following theorem on the difference between predictors.

**Theorem B.6.** *Let $f_{\theta_1}, f_{\theta_2} \in \mathcal{F}_\Theta$ two predictors with respective parameters $\theta_1 = (\psi_1, \mathbf{U}_1, v_1, \Phi_1)$ and $\theta_2 = (\psi_2, \mathbf{U}_2, v_2, \Phi_2)$. We have that*

$$|f_{\theta_1}(x) - f_{\theta_2}(x)|$$

*is upper bounded by*

$$M_\Theta \left\| \Phi_1 - \Phi_2 \right\| + C_\mathbf{A} \sum_{j=1}^q \left\| \mathbf{A}_j^1 - \mathbf{A}_j^2 \right\| + C_\mathbf{b} \sum_{j=1}^q \left\| \mathbf{b}_j^1 - \mathbf{b}_j^2 \right\| + C_\mathbf{U} \left\| \mathbf{U}_1 - \mathbf{U}_2 \right\| + C_v \left\| v_1 - v_2 \right\|.$$

*In turn,*

$$|f_{\theta_1}(\mathbf{x}^D) - f_{\theta_2}(\mathbf{x}^D)|$$

*is upper bounded by*

$$M_\Theta^D \left\| \Phi_1 - \Phi_2 \right\| + C_\mathbf{A}^D \sum_{j=1}^q \left\| \mathbf{A}_j^1 - \mathbf{A}_j^2 \right\| + C_\mathbf{b}^D \sum_{j=1}^q \left\| \mathbf{b}_j^1 - \mathbf{b}_j^2 \right\| + C_\mathbf{U} \left\| \mathbf{U}_1 - \mathbf{U}_2 \right\| + C_v \left\| v_1 - v_2 \right\|.$$

*The Lipschitz constants are given explicitly in the proof.*

The proof is in Appendix D.7.

### B.4    PROOF OF PROPOSITION 4.2 (COVERING NUMBER)

We follow the proof strategy of Chen et al. (2020a). Starting from Theorem B.6, one can see that since

$$\sup_x \| f_{\theta_1}(x) - f_{\theta_2}(x) \|$$

$$\leq M_\Theta \left\| \Phi_1 - \Phi_2 \right\| + C_\mathbf{A} \sum_{j=1}^q \left\| \mathbf{A}_j^1 - \mathbf{A}_j^2 \right\| + C_\mathbf{b} \sum_{j=1}^q \left\| \mathbf{b}_j^1 - \mathbf{b}_j^2 \right\| + C_\mathbf{U} \left\| \mathbf{U}_1 - \mathbf{U}_2 \right\| + C_v \left\| v_1 - v_2 \right\|,$$

where the supremum is taken on all $x$ such that $\|x\|_{\text{1-var},[0,1]} \leq L_x$ and $\|x_0\| \leq B_x$, it is sufficient to have $\hat{\theta} = (\hat{\Phi}, \hat{\psi}, \hat{\mathbf{U}}, \hat{v})$ such that for all $j = 1, \ldots, q$

$$\left\| \mathbf{A}_j - \hat{\mathbf{A}}_j \right\| \leq \frac{\eta}{(2q+3)C_\mathbf{A}}, \quad \left\| \mathbf{b}_j - \hat{\mathbf{b}}_j \right\| \leq \frac{\eta}{(2q+3)C_\mathbf{b}}, \tag{13}$$

and

$$\left\| \Phi - \hat{\Phi} \right\| \leq \frac{\eta}{(2q+3)M_\Theta}, \quad \left\| \mathbf{U} - \hat{\mathbf{U}} \right\| \leq \frac{\eta}{(2q+3)C_\mathbf{U}}, \quad \|v - \hat{v}\| \leq \frac{\eta}{(2q+3)C_v} \tag{14}$$

to obtain an $\eta$ covering of $\mathcal{F}_\Theta$, since in this case we get that

$$\sup_x \left\| f_{\theta_1}(x) - f_{\hat{\theta}}(x) \right\| \leq \eta. \tag{15}$$

We now use Lemma A.9 and denote by $\mathcal{G}_\Phi = \{h : z \mapsto \Phi^\top z, \|\Phi\| \leq B_\Phi\}$, and use the corresponding definitions for $\mathcal{G}_{\mathbf{A}_j}, \mathcal{G}_{\mathbf{b}_j}, \mathcal{G}_\mathbf{U}$ and $\mathcal{G}_v$. We first get that

$$\mathcal{N}\left(\mathcal{G}_\Phi, \frac{\eta}{(2q+3)M_\Theta}\right) \leq \left(1 + \frac{(4q+6)B_\Phi M_\Theta}{\eta}\right)^p, \tag{16}$$

$$\mathcal{N}\left(\mathcal{G}_\mathbf{U}, \frac{\eta}{(2q+3)C_\mathbf{U}}\right) \leq \left(1 + \frac{(4q+6)\min\{\sqrt{p}, \sqrt{d}\}B_\mathbf{U} C_\mathbf{U}}{\eta}\right)^{dp} \quad \text{and} \tag{17}$$

$$\mathcal{N}\left(\mathcal{G}_v, \frac{\eta}{(2q+3)C_v}\right) \leq \left(1 + \frac{(4q+6)B_v C_v}{\eta}\right)^p. \tag{18}$$

Turning to $\psi$, for all $j = 1, \ldots, q-1$

$$\mathcal{N}\left(\mathcal{G}_{\mathbf{A}_j}, \frac{\eta}{(2q+3)C_\mathbf{A}}\right) \leq \left(1 + \frac{(4q+6)\sqrt{p}B_\mathbf{A} C_\mathbf{A}}{\eta}\right)^{p^2} \tag{19}$$

$$\mathcal{N}\left(\mathcal{G}_{\mathbf{b}_j}, \frac{\eta}{(2q+3)C_\mathbf{b}}\right) \leq \left(1 + \frac{(4q+6)\sqrt{p}B_\mathbf{b} C_\mathbf{b}}{\eta}\right)^p, \tag{20}$$

$$\tag{21}$$

and finally, since $A_q$ and $b_q$ do not have the same dimension as the previous weights and biases,

$$\mathcal{N}\Big(\mathcal{G}_{\mathbf{A}_q}, \frac{\eta}{(2q+3)C_{\mathbf{A}}}\Big) \leq \Big(1 + \frac{(4q+6)\sqrt{dp}B_{\mathbf{A}}C_{\mathbf{A}}}{\eta}\Big)^{dp^2}, \tag{22}$$

$$\mathcal{N}\Big(\mathcal{G}_{\mathbf{b}_q}, \frac{\eta}{(2q+3)C_{\mathbf{b}}}\Big) \leq \Big(1 + \frac{(4q+6)\min\{\sqrt{d}, \sqrt{p}\}B_{\mathbf{b}}C_{\mathbf{b}}}{\eta}\Big)^{dp} \tag{23}$$

The covering number of $\mathcal{F}_{\Theta}$ is obtained by multiplying the covering number of each functional class (Chen et al., 2020a). Defining

$$K_1 := \max\{B_{\Phi}M_{\Theta}, B_v C_v\}, \tag{24}$$
$$K_2 := \max\{B_{\mathbf{b}}C_{\mathbf{b}}, B_{\mathbf{A}}C_{\mathbf{A}}, B_{\mathbf{U}}C_{\mathbf{U}}\} \tag{25}$$
$$\tag{26}$$

and using the fact that $\min\{\sqrt{d}, \sqrt{p}\} \leq \sqrt{dp}$, we finally get that $\mathcal{N}(\mathcal{F}_{\Theta}, \eta)$ is upper bounded by

$$\Big(1 + \frac{(4q+6)K_1}{\eta}\Big)^{2p} \times \Big(1 + \frac{(4q+6)K_2}{\eta}\sqrt{p}\Big)^{(q-1)p(1+p)} \tag{27}$$

$$\times \Big(1 + \frac{(4q+6)K_2}{\eta}\sqrt{dp}\Big)^{2dp} \times \Big(1 + \frac{(4q+6)K_2}{\eta}\sqrt{dp}\Big)^{dp^2} \tag{28}$$

which is itself bounded by

$$\Big(1 + \frac{(4q+6)K_1}{\eta}\Big)^{2p} \times \Big(1 + \frac{(4q+6)K_2}{\eta}\sqrt{p}\Big)^{(q-1)p(1+p)} \times \Big(1 + \frac{(4q+6)K_2}{\eta}\sqrt{dp}\Big)^{dp(2+p)} \tag{29}$$

The proof is identical for inputs $\mathbf{x}^D$, but with constants

$$K_1^D := \max\{B_{\Phi}M_{\Theta}^D, B_v C_v\}, \tag{30}$$
$$K_2^D := \max\{B_{\mathbf{b}}C_{\mathbf{b}}^D, B_{\mathbf{A}}C_{\mathbf{A}}^D, B_{\mathbf{U}}C_{\mathbf{U}}\} \tag{31}$$

where the constant $M_{\Theta}^D$ is given in Lemma 3.3, the constants $\mathbf{B}_A, \mathbf{B}_b, \mathbf{B}_U$ and $\mathbf{B}_v$ are given in Section 3.1 and the remaining constants come from the proof provided in D.7. This concludes the proof.

## B.5 Proof of Proposition 4.2 (Rademacher Complexity)

We apply Lemma A.7 to the class $\mathcal{F}_{\Theta}$. We trivially have that $\mathbf{0} \in \mathcal{F}$, since this function is recovered by taking $\theta = \mathbf{0}$. By Lemma 3.3, the value of $f_\theta$ is bounded by

$$M_{\Theta} = B_{\Phi}L_{\sigma} \exp\big((B_{\mathbf{A}}L_{\sigma})^q L_x\big)\Big(B_{\mathbf{U}}B_x + B_v + \kappa_{\Theta}(\mathbf{0})L_x\Big). \tag{32}$$

In our setup, we get, from Proposition 4.2, that

$$\int_{\eta}^{2M_{\Theta}\sqrt{n}} \sqrt{\log \mathcal{N}(\mathcal{F}_{\Theta}, s)} ds \tag{33}$$

$$\leq \int_{\eta}^{2M_{\Theta}\sqrt{n}} \left[ 2p \log \left(1 + \frac{(4q+6)K_1}{s}\right) + (q-1)p(p+1) \log \left(1 + \frac{(4q+6)\sqrt{p}K_2}{s}\right) \right. \tag{34}$$

$$\left. + dp(2+p) \log \left(1 + \frac{(4q+6)\sqrt{dp}K_1}{s}\right) \right]^{1/2} ds \tag{35}$$

$$\leq 2M_{\Theta}\sqrt{n} \left[ 2p \log \left(1 + \frac{(4q+6)K_1}{\eta}\right) + (q-1)p(p+1) \log \left(1 + \frac{(4q+6)\sqrt{p}K_2}{\eta}\right) \right. \tag{36}$$

$$\left. + dp(2+p) \log \left(1 + \frac{(4q+6)\sqrt{dp}K_2}{\eta}\right) \right]^{\frac{1}{2}}. \tag{37}$$

Since for $x > 1$, $\log(1 + x) \leq \log(2x)$, we have, for $\eta$ small enough that

$$\int_{\eta}^{2M_{\Theta}\sqrt{n}} \sqrt{\log \mathcal{N}(\mathcal{F}_{\Theta}, s)} ds \tag{38}$$

$$\leq 2M_{\Theta}\sqrt{n} \left[ 2p \log \frac{(8q+12)K_1}{\eta} + (q-1)p(p+1) \log \frac{(8q+12)\sqrt{p}K_2}{\eta} \right. \tag{39}$$

$$\left. + dp(2+p) \log \frac{(8q+12)\sqrt{dp}K_2}{\eta} \right]^{\frac{1}{2}} \tag{40}$$

Taking $\eta = \frac{1}{\sqrt{n}}$, one gets that $\text{Rad}(\mathcal{F}_{\Theta})$ is upper bounded by

$$\frac{4}{n} + \frac{24M_{\Theta}}{\sqrt{n}} \left[ 2p \log \left(C_q \sqrt{n} K_1\right) + (q-1)p(p+1) \log \left(C_q \sqrt{np} K_2\right) + dp(2+p) \log \left(C_q \sqrt{ndp} K_2\right) \right]^{\frac{1}{2}}$$

with $C_q = (8q + 12)$.

## B.6 PROOF OF THEOREM 4.1

We can direcltly apply the results from Mohri et al. (2018), Theorem 11.3. Since our loss is $L_{\ell}$-Lipschitz and bounded by $M_{\ell}$, this gets us that with probability at least $1 - \delta$,

$$\mathcal{R}^D(\hat{f}^D) \leq \mathcal{R}_n(\hat{f}^D) + \frac{24M_{\Theta}^D L_{\ell}}{\sqrt{n}} \sqrt{2p U_1^D + (q-1)p(p+1)U_2^D + dp(2+p)U_3^D} + M_{\ell} \sqrt{\frac{\log 1/\delta}{2n}} \tag{41}$$

with $U_1^D := \log C_q \sqrt{n} K_1^D$, $U_2^D := \log C_q \sqrt{np} K_2^D$ and $U_3^D := \log C_q \sqrt{ndp} K_2^D$.

# C PROOF OF BIAS BOUNDS

## C.1 PROOF OF LEMMA 5.3

For all $f \in \mathcal{F}_\Theta$, we have the decomposition

$$\mathcal{R}^D(\hat{f}^D) - \mathcal{R}(\mathbf{f}^*) = \mathcal{R}^D(\hat{f}^D) - \mathcal{R}_n^D(\hat{f}^D) \tag{42}$$
$$+ \mathcal{R}_n^D(\hat{f}^D) - \mathcal{R}_n^D(f) \tag{43}$$
$$+ \mathcal{R}_n^D(f) - \mathcal{R}_n(f) \tag{44}$$
$$+ \mathcal{R}_n(f) - \mathcal{R}(f) \tag{45}$$
$$+ \mathcal{R}(f) - \mathcal{R}(\mathbf{f}^\star). \tag{46}$$

By optimality of $\hat{f}^D$, we have that almost surely

$$\mathcal{R}_n^D(\hat{f}^D) - \mathcal{R}_n^D(f) \le 0. \tag{47}$$

One is then left with the inequality

$$\mathcal{R}^D(\hat{f}^D) - \mathcal{R}(\mathbf{f}^*) \le \mathcal{R}^D(\hat{f}^D) - \mathcal{R}_n^D(\hat{f}^D) \tag{48}$$
$$+ \mathcal{R}_n^D(f) - \mathcal{R}_n(f) \tag{49}$$
$$+ \mathcal{R}_n(f) - \mathcal{R}(f) \tag{50}$$
$$+ \mathcal{R}(f) - \mathcal{R}(\mathbf{f}^\star). \tag{51}$$

Taking the supremum on the two differences between the empirical and expected risk yields that

$$\mathcal{R}^D(\hat{f}^D) - \mathcal{R}(\mathbf{f}^*) \le \sup_{g \in \mathcal{F}_\Theta} \left[ \mathcal{R}^D(g) - \mathcal{R}_n^D(g) \right] + \sup_{g \in \mathcal{F}_\Theta} \left[ \mathcal{R}(g) - \mathcal{R}_n(g) \right] \tag{52}$$
$$+ \mathcal{R}_n^D(f) - \mathcal{R}_n(f) + \mathcal{R}(f) - \mathcal{R}(\mathbf{f}^\star). \tag{53}$$

This concludes the proof.

## C.2 BOUNDING THE DISCRETIZATION BIAS

We have the following bound of the discretization bias.

**Proposition C.1.** *For any $f_\theta \in \mathcal{F}_\Theta$, we have that*

$$\mathcal{R}_n^D(f_\theta) - \mathcal{R}_n(f_\theta) \le C_\Theta^1 |D|,$$

*where $C_\Theta^1$ is equal to*

$$L_\ell B_\Phi L_x \left[ 1 + \left( L_\sigma B_\mathbf{A} \right)^q \left( \left( L_\sigma B_\mathbf{A} \right)^q \frac{\min\{M_\Theta^D, M_\Theta\}}{B_\Phi} + \kappa_\Theta(\mathbf{0}) \right) L_x \right].$$

*Proof.* Take $f_\theta \in \mathcal{F}_\Theta$. We have

$$\mathcal{R}_n^D(f_\theta) - \mathcal{R}_n(f_\theta) = \frac{1}{n} \sum_{i=1}^n \left[ \ell(y^i, f_\theta(\mathbf{x}^{D,i})) - \ell(y^i, f_\theta(x^i)) \right]. \tag{54}$$

Considering a single individual $i \in \{1, \ldots, n\}$, we have

$$\ell(y^i, f_\theta(\mathbf{x}^{D,i})) - \ell(y^i, f_\theta(x^i)) \le L_\ell |f_\theta(\mathbf{x}^{D,i}) - f_\theta(x^i)| \tag{55}$$

using the Lipschitz continuity of the loss function with respect to its second argument. From the Cauchy-Schwarz inequality, it follows that

$$|f_\theta(\mathbf{x}^{D,i}) - f_\theta(x^i)| = |\Phi^\top (z_1^{D,i} - z_1^i)| \le B_\Phi \left\| z_1^{D,i} - z_1^i \right\| \tag{56}$$

where $z^{D,i}$ and $z_1^i$ refer to the endpoint of the latent space trajectory of the NCDE, resp. with discrete and continuous input, associated to the predictor $f_\theta$.

$z_1^{D,i} = f_\theta(\mathbf{x}^{D,i})$ and $z_1^i = f_\theta(x^i)$ correspond to the endpoint of two CDEs with identical vector field and identical initial condition, but whose driving paths differ.

We are now ready to use the continuity of the flow stated in Theorem B.5. $z_1^{D,i} = f_\theta(\mathbf{x}^{D,i})$ and $z_1^i = f_\theta(x^i)$ correspond to the endpoint of two CDEs with identical vector field and identical initial condition, but whose driving paths differ. Theorem B.5 thus collapses to

$$\left\| z_1^{D,i} - z_1^i \right\| \le \exp\left( L_{\mathbf{G}_\psi} L_x \right) \left( 1 + L_{\mathbf{G}_\psi} L_x C_1(\mathbf{G}_\psi) \right) \left\| x^i - \tilde{\mathbf{x}}^i \right\|_{\infty, [0,1]}, \tag{57}$$

where $L_{\mathbf{G}_\psi}$ is the Lipschitz constant of the neural vector field $\mathbf{G}_\psi$. Since, according to Lemmas B.1 and B.2,

$$L_{\mathbf{G}_\psi} = \left( L_\sigma B_\mathbf{A} \right)^q$$

and

$$C_1(\mathbf{G}_\psi) = \left[ \left( L_\sigma B_\mathbf{A} \right)^q \frac{1}{B_\Phi} \min\{M_\Theta, M_\Theta^D\} + \kappa_\Theta(\mathbf{0}) \right]$$

we obtain the inequality

$$\left\| z_1^{D,i} - z_1^i \right\| \tag{58}$$

$$\le \exp\left( (L_\sigma B_\mathbf{A})^q L_x \right) \left\| x - \tilde{\mathbf{x}}^i \right\|_{\infty, [0,1]} \left[ 1 + \left( L_\sigma B_\mathbf{A} \right)^q \left( \left( L_\sigma B_\mathbf{A} \right)^q \frac{\min\{M_\Theta^D, M_\Theta\}}{B_\Phi} + \kappa_\Theta(\mathbf{0}) \right) L_x \right] \tag{59}$$

We also have that

$$\left\| \mathbf{x}^{D,i} - x^i \right\|_{\infty, [0,1]} = \max_{j=1,\dots,K} \max_{s \in [t_j, t_{j+1}]} \left\| x_{t_j}^i - x_s^i \right\| \le L_x |D| \tag{60}$$

since the discretization is identical between individuals. Putting everything together, this yields for all $i = 1, \dots, n$ that

$$|f_\theta(\mathbf{x}^{D,i}) - f_\theta(x^i)| \tag{61}$$

$$\le B_\Phi \exp\left( (L_\sigma B_\mathbf{A})^q L_x \right) L_x \left[ 1 + \left( L_\sigma B_\mathbf{A} \right)^q \left( \left( L_\sigma B_\mathbf{A} \right)^q \frac{\min\{M_\Theta^D, M_\Theta\}}{B_\Phi} + \kappa_\Theta(\mathbf{0}) \right) L_x \right] |D| \tag{62}$$

and finally

$$\mathcal{R}_n^D(f_\theta) - \mathcal{R}_n(f_\theta) \tag{63}$$

$$\le L_\ell B_\Phi \exp\left( (L_\sigma B_\mathbf{A})^q L_x \right) L_x \left[ 1 + \left( L_\sigma B_\mathbf{A} \right)^q \left( \left( L_\sigma B_\mathbf{A} \right)^q \frac{\min\{M_\Theta^D, M_\Theta\}}{B_\Phi} + \kappa_\Theta(\mathbf{0}) \right) L_x \right] |D| \tag{64}$$

$$=: C_\Theta^1 |D|. \tag{65}$$

This concludes the proof. $\qquad \square$

### C.3 BOUNDING THE APPROXIMATION BIAS

We have the following bound of the approximation bias.

**Proposition C.2.** *For any $f_\theta \in \mathcal{F}_\Theta$ with parameters $\theta = (\Phi, \psi, \mathbf{U}, v)$ the approximation bias $\mathcal{R}(f_\theta) - \mathcal{R}(\mathbf{f}^*)$ is bounded from above by*

$$L_\ell B_\Phi \exp(L_{\mathbf{G}^\star} L_x) L_x \left\| \mathbf{G}_\psi - \mathbf{G}^\star \right\|_{\infty, \Omega} + L_\ell B_\Phi \exp(L_{\mathbf{G}^\star} L_x) \left\| \varphi^\star - NN_{\mathbf{U},v} \right\|_{\infty, B_x}$$

$$+ \frac{L_\ell M_\Theta}{B_\Phi} \left\| \Phi_\star - \Phi \right\|,$$

*where*

$$\Omega = \left\{ u \in \mathbb{R}^p \mid \|u\| \le (L_\sigma(B_\mathbf{U} B_x + B_v) + L_\sigma B_\mathbf{b} L_x) \exp(L_\sigma B_\mathbf{A} L_x) \right\}.$$

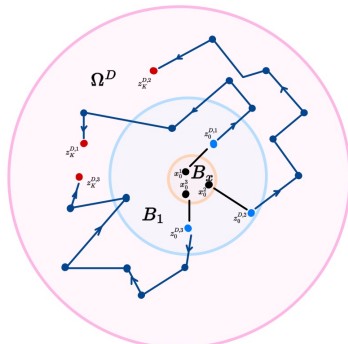

Figure 5: A schematic illustration of the main bounded sets used in the proof. The initial values of the time series used for initializing the NCDE lie in the orange ball of diameter $B_x$. The value $z_0$ then lies in the blue ball $B_1$, whose diameter is upper bounded by a function of $B_{\mathbf{U}}, B_v, B_x$ and $L_\sigma$. Finally, the trajectories evolves during time but stay within the ball $\Omega^D$.

*Proof.* For any $f_\theta \in \mathcal{F}_\Theta$ with continuous input, the approximation bias writes

$$\mathcal{R}(f_\theta) - \mathcal{R}(\mathbf{f}^\star) = \mathbb{E}\Big[\ell(y, f_\theta(x)) - \ell(y, \mathbf{f}^\star(x))\Big]. \tag{66}$$

Using the Lipschitz continuity of $\ell$ and the Cauchy-Schwarz inequality, one gets

$$\ell(y, f_\theta(x)) - \ell(y, \mathbf{f}^\star(x)) \leq L_\ell \|\Phi_\star - \Phi\| \|z_1\| + L_\ell B_\Phi \|z_1^\star - z_1\|. \tag{67}$$

Since $z_1^\star$ and $z_1$ are the solution to two CDEs with different vector fields and different initial conditions, the continuity of the flow stated in Theorem B.5 yields

$$\|z_1^\star - z_1\| \leq \exp(L_{\mathbf{G}^\star} L_x)\Big[L \max_{u \in \Omega} \|\mathbf{G}_\psi(u) - \mathbf{G}^\star(u)\| + \max_{\|u\| \leq B_x} \|\varphi^\star(u) - \mathrm{NN}_{\mathbf{U},v}(u)\|\Big] \tag{68}$$

where

$$\Omega \subset \big\{u \in \mathbb{R}^p \mid \|u\| \leq \frac{1}{B_\Phi} M_\Theta\big\}$$

Putting everything together and taking expectations yields

$$\mathcal{R}(f_\theta) - \mathcal{R}(\mathbf{f}^\star) \leq L_\ell B_\Phi \exp(L_{\mathbf{G}^\star} L_x)\Big[L_x \max_{u \in \Omega} \|\mathbf{G}_\psi(u) - \mathbf{G}^\star(u)\| \tag{69}$$

$$+ \max_{\|u\| \leq B_x} \|\varphi^\star(u) - \mathrm{NN}_{\mathbf{U},v}(u)\|\Big] \tag{70}$$

$$+ \frac{L_\ell M_\Theta}{B_\Phi} \|\Phi_\star - \Phi\|. \tag{71}$$

This concludes the proof.

$\square$

## C.4 PROOF OF THEOREM 5.4

We gave bounds for the discretization and approximation biases in Propositions C.1 and C.2. We now have to bound $\sup\limits_{g \in \mathcal{F}_\Theta} \Big[\mathcal{R}^D(g) - \mathcal{R}_n^D(g)\Big]$ and $\sup\limits_{g \in \mathcal{F}_\Theta} \Big[\mathcal{R}(g) - \mathcal{R}_n(g)\Big]$ to obtain a bound for the

total risk via the total risk decomposition given in Lemma 5.3. It is clear that for any predictor $f_\theta$ parameterized by $\theta = (\Phi, \psi, \mathbf{U}, v)$ we have

$$\mathcal{R}^D(\hat{f}^D) - \mathcal{R}(\mathbf{f}^*) \tag{72}$$

$$\leq \sup_{g \in \mathcal{F}_\Theta} \left[ \mathcal{R}^D(g) - \mathcal{R}_n^D(g) \right] + \sup_{g \in \mathcal{F}_\Theta} \left[ \mathcal{R}(g) - \mathcal{R}_n(g) \right] \tag{73}$$

$$+ L_\ell C_\Theta^1 |D| \tag{74}$$

$$+ L_\ell B_\Phi \exp(L_{\mathbf{G}^\star} L_x) \left[ L_x \left\| \mathbf{G}_\psi - \mathbf{G}^\star \right\|_{\infty, \Omega} + \left\| \varphi^\star - \mathrm{NN}_{\mathbf{U}, v} \right\|_{\infty, B_x} \right] \tag{75}$$

$$+ \frac{L_\ell M_\Theta}{B_\Phi} \left\| \Phi_\star - \Phi \right\|. \tag{76}$$

Now, by a classical symmetrization argument - see for instance Bach (2021), Proposition 4.2 - the obtained bounds on the Rademacher complexity imply that

$$\sup_{g \in \mathcal{F}_\Theta} \left[ \mathcal{R}(g) - \mathcal{R}_n(g) \right] \leq 2\mathrm{Rad}(\mathcal{F}_\Theta) \tag{77}$$

and

$$\sup_{g \in \mathcal{F}_\Theta^D} \left[ \mathcal{R}^D(g) - \mathcal{R}_n^D(g) \right] \leq 2\mathrm{Rad}(\mathcal{F}_\Theta^D). \tag{78}$$

This gives

$$\sup_{g \in \mathcal{F}_\Theta} \left[ \mathcal{R}^D(g) - \mathcal{R}_n^D(g) \right] + \sup_{g \in \mathcal{F}_\Theta} \left[ \mathcal{R}(g) - \mathcal{R}_n(g) \right] \leq 4 \max\{\mathrm{Rad}(\mathcal{F}_\Theta^D), \mathrm{Rad}(\mathcal{F}_\Theta)\} \tag{79}$$

To finish the proof, notice that since the risk decomposition

$$\mathcal{R}^D(\hat{f}^D) - \mathcal{R}(\mathbf{f}^*) \leq \sup_{g \in \mathcal{F}_\Theta} \left[ \mathcal{R}^D(g) - \mathcal{R}_n^D(g) \right] + \sup_{g \in \mathcal{F}_\Theta} \left[ \mathcal{R}(g) - \mathcal{R}_n(g) \right]$$
$$+ \mathcal{R}_n^D(f) - \mathcal{R}_n(f) + \mathcal{R}(f) - \mathcal{R}(\mathbf{f}^\star)$$
$$\leq 4 \max\{\mathrm{Rad}(\mathcal{F}_\Theta^D), \mathrm{Rad}(\mathcal{F}_\Theta)\} + C_\Theta^1 |D|$$
$$+ L_\ell B_\Phi \exp(L_{\mathbf{G}^\star} L_x) \left[ L_x \left\| \mathbf{G}_\psi - \mathbf{G}^\star \right\|_{\infty, \Omega} + \left\| \varphi^\star - \mathrm{NN}_{\mathbf{U}, v} \right\|_{\infty, B_x} \right]$$
$$+ \frac{L_\ell M_\Theta}{B_\Phi} \left\| \Phi_\star - \Phi \right\|$$

holds for every $f_\theta \in \mathcal{F}_\Theta$ parameterized by $(\Phi, \psi, \mathbf{U}, v)$, one may take the infimum over all possible predictors in $\mathcal{F}_\Theta$. Since this parameter space is compact by definition, the infimum is a minimum and

$$\mathcal{R}^D(\hat{f}^D) - \mathcal{R}(\mathbf{f}^*)$$
$$\leq 4 \max\{\mathrm{Rad}(\mathcal{F}_\Theta^D), \mathrm{Rad}(\mathcal{F}_\Theta)\} + C_\Theta^1 |D|$$
$$+ L_\ell B_\Phi \exp(L_{\mathbf{G}^\star} L_x) \left[ L_x \min_\psi \left\| \mathbf{G}_\psi - \mathbf{G}^\star \right\|_{\infty, \Omega} + \min_{\mathbf{U}, v} \left\| \varphi^\star - \mathrm{NN}_{\mathbf{U}, v} \right\|_{\infty, B_x} \right]$$
$$+ \frac{L_\ell M_\Theta}{B_\Phi} \underbrace{\min_\Phi \left\| \Phi_\star - \Phi \right\|}_{=0}$$

which concludes the proof.

# D SUPPLEMENTARY PROOFS

## D.1 PROOF OF LEMMA B.1

Take $\mathbf{G}_\psi : \mathbb{R}^p \to \mathbb{R}^{p \times d}$ a neural vector field and $z, w \in \mathbb{R}^p$. Define

$$\mu_h(z) := \sigma\Big(\mathbf{A}_h \sigma\Big(\mathbf{A}_{h-1}\sigma\Big(\ldots\sigma\Big(\mathbf{A}_1 z + \mathbf{b}_1\Big)\Big) + \mathbf{b}_{h-1}\Big) + \mathbf{b}_h\Big)$$

for all $h = 2, \ldots, q$. One directly has

$$\|\mathbf{G}_\psi(z) - \mathbf{G}_\psi(w)\| = \|\mu_q(z) - \mu_q(w)\| \leq L_\sigma \left\|\mathbf{A}_q\big(\mu_{q-1}(z) - \mu_{q-1}(w)\big)\right\| \tag{80}$$

$$\leq L_\sigma B_\mathbf{A} \left\|\mu_{q-1}(z) - \mu_{q-1}(w)\right\| \tag{81}$$

$$\leq \ldots \tag{82}$$

$$\leq \Big(L_\sigma B_\mathbf{A}\Big)^q \|z - w\|. \tag{83}$$

This proves our claim.

## D.2 PROOF OF LEMMA B.2

Let $z \in \mathbb{R}^p$ and define

$$\mu_h(z) := \sigma\Big(\mathbf{A}_h^1 \sigma\Big(\mathbf{A}_{h-1}^1\sigma\Big(\ldots\sigma\Big(\mathbf{A}_1^1 z + \mathbf{b}_1^1\Big)\Big) + \mathbf{b}_{h-1}^1\Big) + \mathbf{b}_h^1\Big)$$

for all $h = 2, \ldots, q$. We have

$$\|\mathbf{G}_\psi(z)\|_{\mathrm{op}} = \|\mu_q(z)\|_{\mathrm{op}} \tag{84}$$

$$\leq \|\mu_q(z) - \mu_q(\mathbf{0}) + \mu_q(\mathbf{0})\| \tag{85}$$

$$\leq \|\mu_q(z) - \mu_q(\mathbf{0})\| + \|\mu_q(\mathbf{0})\|. \tag{86}$$

Using Lemma B.1, we have that

$$\|\mu_q(z) - \mu_q(\mathbf{0})\| \leq \Big(L_\sigma B_\mathbf{A}\Big)^q \|z\|. \tag{87}$$

To finish the proof, we have to bound $\|\mu_q(\mathbf{0})\|$. Since

$$\|\mu_q(\mathbf{0})\| = \|\mu_q(\mathbf{0}) - \sigma(\mathbf{0})\| \leq L_\sigma \left\|\mathbf{A}_q \mu_{q-1}(\mathbf{0})\right\| + L_\sigma B_\mathbf{b} \tag{88}$$

$$\leq L_\sigma B_\mathbf{A} \left\|\mu_{q-1}(\mathbf{0})\right\| + L_\sigma B_\mathbf{b}, \tag{89}$$

using Lemma A.6 yields

$$\|\mu_q(\mathbf{0})\| \leq \big(L_\sigma B_\mathbf{A}\big)^{q-1} \|\sigma(\mathbf{b}_1)\| + L_\sigma B_\mathbf{b} \sum_{j=0}^{q-2} \big(L_\sigma B_\mathbf{A}\big)^j \tag{90}$$

$$\leq L_\sigma B_\mathbf{b} \sum_{j=0}^{q-1} \big(L_\sigma B_\mathbf{A}\big)^j \tag{91}$$

$$= L_\sigma B_\mathbf{b} \frac{\big(L_\sigma B_\mathbf{A}\big)^q - 1}{L_\sigma B_\mathbf{A} - 1}. \tag{92}$$

Combining everything yields

$$\|\mathbf{G}_\psi(z)\|_{\mathrm{op}} \leq \Big(L_\sigma B_\mathbf{A}\Big)^q \|z\| + L_\sigma B_\mathbf{b} \frac{\big(L_\sigma B_\mathbf{A}\big)^q - 1}{L_\sigma B_\mathbf{A} - 1}. \tag{93}$$

## D.3 Proof of Lemma B.3

Define

$$\mu_h^1(z) := \sigma\Big(\mathbf{A}_h^1\sigma\Big(\mathbf{A}_{h-1}^1\sigma\Big(\dots\sigma\Big(\mathbf{A}_1^1 z + \mathbf{b}_1^1\Big)\Big) + \mathbf{b}_{h-1}^1\Big) + \mathbf{b}_h^1\Big)$$

and

$$\mu_h^2(z) := \sigma\Big(\mathbf{A}_h^2\sigma\Big(\mathbf{A}_{h-1}^2\sigma\Big(\dots\sigma\Big(\mathbf{A}_1^2 z + \mathbf{b}_1^2\Big)\Big) + \mathbf{b}_{h-1}^2\Big) + \mathbf{b}_h^2\Big)$$

for all $h = 2\dots, q$. The indice $h$ therefore refers to the *depth* of the neural vector field.

For the first hidden layer, we have

$$\left\|\sigma\Big(\mathbf{A}_1^1 z + \mathbf{b}_1^1\Big) - \sigma\Big(\mathbf{A}_1^2 z + \mathbf{b}_1^2\Big)\right\| \le L_\sigma \left\|\mathbf{b}_1^1 - \mathbf{b}_1^2\right\| + L_\sigma \left\|\mathbf{A}_1^1 z - \mathbf{A}_1^2 z\right\| \tag{94}$$

$$\le L_\sigma \left\|\mathbf{b}_1^1 - \mathbf{b}_1^2\right\| + L_\sigma \left\|\mathbf{A}_1^1 - \mathbf{A}_1^2\right\| \left\|z\right\|. \tag{95}$$

Similarly, for any $h = 2, \dots, q$,

$$\left\|\mu_h^1(z) - \mu_h^2(z)\right\| = \left\|\sigma\big(\mathbf{A}_h^1\mu_{h-1}^1(z) + \mathbf{b}_h^1\big) - \sigma\big(\mathbf{A}_h^2\mu_{h-1}^2(z) + \mathbf{b}_h^2\big)\right\| \tag{96}$$

$$\le L_\sigma \left\|\mathbf{A}_h^1\mu_{h-1}^1(z) - \mathbf{A}_h^2\mu_{h-1}^2(z)\right\| + L_\sigma \left\|\mathbf{b}_h^1 - \mathbf{b}_h^2\right\| \tag{97}$$

$$\le L_\sigma \left\|\Big(\mathbf{A}_h^1 - \mathbf{A}_h^2 + \mathbf{A}_h^2\Big)\mu_{h-1}^1(z) - \mathbf{A}_h^2\mu_{h-1}^2(z)\right\| + L_\sigma \left\|\mathbf{b}_h^1 - \mathbf{b}_h^2\right\| \tag{98}$$

$$\le L_\sigma \left\|\mathbf{A}_h^1 - \mathbf{A}_h^2\right\| \left\|\mu_{h-1}^1(z)\right\| + L_\sigma \left\|\mathbf{A}_h^2\right\| \left\|\mu_{h-1}^1(z) - \mu_{h-1}^2(z)\right\| \tag{99}$$

$$+ L_\sigma \left\|\mathbf{b}_h^1 - \mathbf{b}_h^2\right\|. \tag{100}$$

Since we have assumed that $\left\|\mathbf{A}_q^2\right\| \le B_\mathbf{A}$, this yields

$$\left\|\mu_h^1(z) - \mu_h^2(z)\right\| \le L_\sigma \left\|\mathbf{A}_h^1 - \mathbf{A}_h^2\right\| \left\|\mu_{h-1}^1(z)\right\| + L_\sigma B_\mathbf{A} \left\|\mu_{h-1}^1(z) - \mu_{h-1}^2(z)\right\| \tag{101}$$

$$+ L_\sigma \left\|\mathbf{b}_h^1 - \mathbf{b}_h^2\right\|. \tag{102}$$

We may now use Lemma A.6 to get that

$$\left\|\mu_q^1(z) - \mu_q^2(z)\right\| \le L_\sigma \sum_{j=2}^q \Big[ \left\|\mathbf{A}_j^1 - \mathbf{A}_j^2\right\| \left\|\mu_{j-1}^1(z)\right\| + \left\|\mathbf{b}_j^1 - \mathbf{b}_j^2\right\| \Big] \big(L_\sigma B_\mathbf{A}\big)^{q-j} \tag{103}$$

$$+ \big(L_\sigma B_\mathbf{A}\big)^{q-1} \left\|\sigma\Big(\mathbf{A}_1^1 z + \mathbf{b}_1^1\Big) - \sigma\Big(\mathbf{A}_1^2 z + \mathbf{b}_1^2\Big)\right\| \tag{104}$$

$$\le L_\sigma \sum_{j=2}^q \Big[ \left\|\mathbf{A}_j^1 - \mathbf{A}_j^2\right\| \left\|\mu_{j-1}^1(z)\right\| + \left\|\mathbf{b}_j^1 - \mathbf{b}_j^2\right\| \Big] \big(L_\sigma B_\mathbf{A}\big)^{q-j} \tag{105}$$

$$+ L_\sigma^q B_\mathbf{A}^{q-1} \left\|\mathbf{b}_1^1 - \mathbf{b}_1^2\right\| + L_\sigma^q B_\mathbf{A}^{q-1} \left\|\mathbf{A}_1^1 - \mathbf{A}_1^2\right\| \left\|z\right\|. \tag{106}$$

We may now use Lemma B.2. For all $h = 1, \dots, q$, we have

$$\left\|\mu_h^1(z)\right\| \le \alpha_h \left\|z\right\| + \beta_h \tag{107}$$

with

$$\alpha_h := \big(L_\sigma B_\mathbf{A}\big)^h \text{ and } \beta_h := L_\sigma B_\mathbf{b}\frac{\big(L_\sigma B_\mathbf{A}\big)^h - 1}{L_\sigma B_\mathbf{A} - 1}. \tag{108}$$

We furthermore let $\alpha_0 := 1$ and $\beta_0 = 0$. One then has

$$\left\|\mu_q^1(z) - \mu_q^2(z)\right\| \tag{109}$$

$$\le L_\sigma \sum_{j=1}^q \Big[ \left\|\mathbf{A}_j^1 - \mathbf{A}_j^2\right\| \big(\alpha_{j-1} \left\|z\right\| + \beta_{j-1}\big) + \left\|\mathbf{b}_j^1 - \mathbf{b}_j^2\right\| \Big] \big(L_\sigma B_\mathbf{A}\big)^{q-j} \tag{110}$$

and thus

$$\left\|\mathbf{G}_{\psi^1}(z) - \mathbf{G}_{\psi^2}(z)\right\|_{\text{op}} \le \sum_{j=1}^q \Big[ C_\mathbf{b}^j \left\|\mathbf{b}_j^1 - \mathbf{b}_j^2\right\| + C_\mathbf{A}^j(z) \left\|\mathbf{A}_j^1 - \mathbf{A}_j^2\right\| \Big] \tag{111}$$

with $C_\mathbf{A}^j(z) := \big(L_\sigma B_\mathbf{A}\big)^{q-j} L_\sigma\big(\alpha_{j-1} \left\|z\right\| + \beta_{j-1}\big)$ and $C_\mathbf{b}^j = \big(L_\sigma B_\mathbf{A}\big)^{q-j} L_\sigma$. This proves our claim.

## D.4 PROOF OF LEMMA 3.3 AND LEMMA 5.2

We recall that $\|W\|_{\mathrm{op}} := \max_{\|x\|=1} \|Wx\|$, and that $\|W\|_{\mathrm{op}} \leq \|W\|$, which means that $\|Wx\| \leq \|W\| \|x\|$ for all $x$. We first prove this result for a general CDE

$$dz_t = \mathbf{F}(z_t)dx_t$$

with initial condition $z_0 \in \mathbb{R}^p$, where the driving path $x$ is supposed to be continuous of bounded variation (or piecewise constant with a finite number of discontinuities.)

By definition,

$$z_t = z_0 + \int_0^t \mathbf{F}(z_t)dx_t. \tag{112}$$

Taking norms, this yields

$$\|z_t\| \leq \|z_0\| + \int_0^t \|\mathbf{F}(z_t)\|_{\mathrm{op}} \|dx_t\|. \tag{113}$$

Notice that since we have assumed $\mathbf{F}$ to be Lipschitz, we have that for all $z \in \mathbb{R}^p$

$$\|\mathbf{F}(z)\|_{\mathrm{op}} \leq \|\mathbf{F}(z) - \mathbf{F}(0)\|_{\mathrm{op}} + \|\mathbf{F}(0)\|_{\mathrm{op}} \tag{114}$$

$$\leq \|\mathbf{F}(z) - \mathbf{F}(0)\| + \|\mathbf{F}(0)\|_{\mathrm{op}} \tag{115}$$

$$\leq L_{\mathbf{F}} \|z\| + \|\mathbf{F}(0)\|_{\mathrm{op}}, \tag{116}$$

where the last inequality follows from the fact that $\mathbf{F}$ is Lipschitz. It follows that

$$\|z_t\| \leq \|z_0\| + \int_0^t (L_{\mathbf{F}} \|z_t\| + \|\mathbf{F}(0)\|_{\mathrm{op}}) \|dx_t\|. \tag{117}$$

Using the fact that $\int_0^t \|dx_s\| = \|x\|_{\text{1-var},[0,t]}$, one gets

$$\|z_t\| \leq \|z_0\| + \|\mathbf{F}(0)\|_{\mathrm{op}} \|x\|_{\text{1-var},[0,t]} + L_{\mathbf{F}} \int_0^t \|z_s\| \|dx_s\|. \tag{118}$$

Applying Gronwall's Lemma for CDEs yields

$$\|z_t\| \leq \left( \|z_0\| + \|\mathbf{F}(0)\|_{\mathrm{op}} \|x\|_{\text{1-var},[0,t]} \right) \exp \left( L_{\mathbf{F}} \|x\|_{\text{1-var},[0,t]} \right). \tag{119}$$

Now, turning to the generative CDE equation 5, we obtain as a consequence that

$$|y| \leq B_\Phi \left( B_\varphi^\star + \|\mathbf{G}^\star(0)\|_{\mathrm{op}} L_x \right) \exp \left( L_{\mathbf{G}^\star} L_x \right) + M_\varepsilon, \tag{120}$$

since $y$ is equal to the sum of a linear transformation of the endpoint of a CDE an a noise term $\varepsilon$ bounded by $M_\varepsilon$. This proves Lemma 5.2.

Turning now to $f_\theta \in \mathcal{F}_\Theta$, we have that

$$|f_\theta(x)| = |\Phi^\top z_1| \leq \|\Phi\| \|z_1\| \leq B_\Phi \|z_1\| \tag{121}$$

and since $z_1$ is the solution of a NCDE, we can directly leverage the previous result to bound $\|z_1\|$, which yields

$$\|z_1\| \leq L_\sigma \exp \left( (B_{\mathbf{A}} L_\sigma)^q L_x \right) \left( B_{\mathbf{U}} B_x + B_v + \kappa_\Theta(\mathbf{0}) L_x \right). \tag{122}$$

As a direct consequence,

$$|f_\theta(x)| \leq B_\Phi L_\sigma \exp \left( (B_{\mathbf{A}} L_\sigma)^q L_x \right) \left( B_{\mathbf{U}} B_x + B_v + \kappa_\Theta(\mathbf{0}) L_x \right) < \infty. \tag{123}$$

We now turn to $f_\theta(\mathbf{x}^D)$. Bounding the value of $z_1^D$ can be done by applying this lemma to the values of $z^D$ at points in $D$, since it is constant between those points. The proof leverages the discrete version of Gronwall's Lemma, stated in Lemma A.5. We have that

$$\|z_{t_k}\| \leq \|z_0\| + \sum_{l=0}^{k-1} \left\|\mathbf{G}_\psi(z_{t_l})\Delta x_{t_{l+1}}\right\| \leq \sum_{l=0}^{k-1} \left\|\mathbf{G}_\psi(z_{t_l})\right\| \left\|\Delta x_{t_{l+1}}\right\|, \tag{124}$$

for all $k = 1, \ldots, K$. Now, since

$$\|\mathbf{G}_\psi(z_{t_l})\| \leq \left(L_\sigma B_\mathbf{A}\right)^q \|z_{t_l}\| + L_\sigma B_\mathbf{b} \frac{\left(L_\sigma B_\mathbf{A}\right)^q - 1}{L_\sigma B_\mathbf{A} - 1} \tag{125}$$

we have

$$\|z_{t_k}\| \leq \|z_0\| + L_\sigma B_\mathbf{b} \frac{\left(L_\sigma B_\mathbf{A}\right)^q - 1}{L_\sigma B_\mathbf{A}} \sum_{l=0}^{k-1} \left\|\Delta x_{t_{l+1}}\right\| + \left(L_\sigma B_\mathbf{A}\right)^q \sum_{l=0}^{k-1} \|z_{t_l}\| \|\Delta x_{l+1}\| \tag{126}$$

$$\leq \|z_0\| + L_\sigma B_\mathbf{b} \frac{\left(L_\sigma B_\mathbf{A}\right)^q - 1}{L_\sigma B_\mathbf{A}} L_x + \left(L_\sigma B_\mathbf{A}\right)^q \sum_{l=0}^{k-1} \|z_{t_l}\| \|\Delta x_{l+1}\| \tag{127}$$

and one gets by the discrete Gronwall Lemma that

$$\left\|z_{t_k}^D\right\| \leq \left(\|z_0\| + L_\sigma B_\mathbf{b} \frac{\left(L_\sigma B_\mathbf{A}\right)^q - 1}{L_\sigma B_\mathbf{A}} L_x\right) \prod_{l=0}^{k-1} \left(1 + \left(L_\sigma B_\mathbf{A}\right)^q \left\|\Delta x_{t_{l+1}}\right\|\right) \tag{128}$$

$$\leq \left(\|z_0\| + L_\sigma B_\mathbf{b} \frac{\left(L_\sigma B_\mathbf{A}\right)^q - 1}{L_\sigma B_\mathbf{A}} L_x\right) \prod_{l=0}^{k-1} \left(1 + \left(L_\sigma B_\mathbf{A}\right)^q L_x|D|\right) \tag{129}$$

since $\left\|\Delta x_{t_{l+1}}\right\| \leq L_x|t_{l+1} - t_l| \leq L_x|D|$, and finally

$$\left\|z_{t_k}^D\right\| \leq \left(L_\sigma B_\mathbf{U} B_x + L_\sigma B_v + L_\sigma B_\mathbf{b} \frac{\left(L_\sigma B_\mathbf{A}\right)^q - 1}{L_\sigma B_\mathbf{A}} L_x\right) \prod_{l=0}^{k-1} \left(1 + \left(L_\sigma B_\mathbf{A}\right)^q L_x|D|\right) \tag{130}$$

In the end, this gets us that

$$|f_\theta(\mathbf{x}^D)| \leq B_\Phi L_\sigma \left(L_\sigma B_\mathbf{U} B_x + B_v + L_\sigma B_\mathbf{b} \frac{\left(L_\sigma B_\mathbf{A}\right)^q - 1}{L_\sigma B_\mathbf{A}} L_x\right) \prod_{l=0}^{K-1} \left(1 + \left(L_\sigma B_\mathbf{A}\right)^q L_x|D|\right) \tag{131}$$

$$= B_\Phi L_\sigma \left(L_\sigma B_\mathbf{U} B_x + B_v + L_\sigma B_\mathbf{b} \frac{\left(L_\sigma B_\mathbf{A}\right)^q - 1}{L_\sigma B_\mathbf{A}} L_x\right) \left(1 + \left(L_\sigma B_\mathbf{A}\right)^q L_x|D|\right)^K. \tag{132}$$

Note that by Gronwall's Lemma, one also has the less tighter bound

$$|f_\theta(\mathbf{x}^D)| \leq B_\Phi L_\sigma \exp\left((B_\mathbf{A} L_\sigma)^q L_x\right) \left(B_\mathbf{U} B_x + B_v + \kappa_\Theta(\mathbf{0}) L_x\right). \tag{133}$$

This concludes the proof.

### D.5   PROOF OF PROPOSITION B.4

We first consider a general CDE

$$dz_t = \mathbf{F}(z_t)dx_t \tag{134}$$

with initial condition $z_0 \in \mathbb{R}^p$. Take $s, r \in [0, t]$. By definition,

$$\|z_r - z_s\| = \left\| \int_s^r \mathbf{F}(z_u) dx_u \right\| \tag{135}$$

$$= \left\| \int_s^r \mathbf{F}(z_u - z_s + z_s) dx_u \right\| \tag{136}$$

$$\leq \int_s^r \left( L_{\mathbf{F}} \|z_u - z_s + z_s\| + \|\mathbf{F}(0)\|_{\mathrm{op}} \right) \|dx_u\| \tag{137}$$

$$\leq \int_s^r \left( L_{\mathbf{F}} \|z_u - z_s\| + L_{\mathbf{F}} \|z_s\| + \|\mathbf{F}(0)\|_{\mathrm{op}} \right) \|dx_u\| \tag{138}$$

$$= \left( L_{\mathbf{F}} \|z_s\| + \|\mathbf{F}(0)\|_{\mathrm{op}} \right) \|x\|_{1\text{-var},[s,r]} + L_{\mathbf{F}} \int_s^r \|z_u - z_s\| \|dx_u\| . \tag{139}$$

Now since $z_s$ is the solution of a CDE evaluated at $s$, it can be bounded by Lemma 3.3 by

$$\|z_s\| \leq \left( \|z_0\| + \|\mathbf{F}(0)\|_{\mathrm{op}} \|x\|_{1\text{-var},[0,s]} \right) \exp(L_{\mathbf{F}} \|x\|_{1\text{-var},[0,s]}). \tag{140}$$

This means that

$$\|z_r - z_s\| \tag{141}$$

$$\leq \left[ L_{\mathbf{F}} \left( \|z_0\| + \|\mathbf{F}(0)\|_{\mathrm{op}} \|x\|_{1\text{-var},[0,s]} \right) \exp(L_{\mathbf{F}} \|x\|_{1\text{-var},[0,s]}) + \|\mathbf{F}(0)\|_{\mathrm{op}} \right] \|x\|_{1\text{-var},[s,r]} \tag{142}$$

$$+ L_{\mathbf{F}} \int_s^r \|z_u - z_s\| \|dx_u\| . \tag{143}$$

We may now use Gronwall's Lemma and the fact that $\|x\|_{1\text{-var},[0,s]} \leq L_x$ for all $s \in [0, 1]$ to obtain that

$$\|z_r - z_s\| \leq \left[ L_{\mathbf{F}} \left( \|z_0\| + \|\mathbf{F}(0)\|_{\mathrm{op}} L_x \right) \exp(L_{\mathbf{F}} L_x) + \|\mathbf{F}(0)\|_{\mathrm{op}} \right] \exp(L_{\mathbf{F}} L_x) \|x\|_{1\text{-var},[s,r]} . \tag{144}$$

and thus

$$\|z_r - z_s\| \leq C_1(\mathbf{F}) \|x\|_{1\text{-var},[s,r]} . \tag{145}$$

This means that the variations of $z$ on arbitrary intervals $[r, s]$ are bounded by the variations of $x$ on the same interval, times an interval independent constant.

From this, we may immediately conclude that

$$\|z\|_{1\text{-var},[0,t]} \leq C_1(\mathbf{F}) \|x\|_{1\text{-var},[0,t]} , \tag{146}$$

which concludes the proof for a general CDE. We now turn to $f_\theta \in \mathcal{F}_\Theta$. The previous result allows to bound the total variation of a CDE and thus of the trajectory of the latent state. Using this proposition with

$$C_1(\mathbf{G}_\psi) := \tag{147}$$

$$\left[ \left( L_\sigma B_{\mathbf{A}} \right)^q \left( L_\sigma B_{\mathbf{U}} B_x + L_\sigma B_v + \kappa_\Theta(\mathbf{0}) L_x \right) \exp \left( (L_\sigma B_{\mathbf{A}})^q L_x \right) + \kappa_\Theta(\mathbf{0}) \right] \exp \left( (L_\sigma B_{\mathbf{A}})^q L_x \right) \tag{148}$$

$$= \left[ \left( L_\sigma B_{\mathbf{A}} \right)^q \frac{1}{B_\Phi} M_\Theta + \kappa_\Theta(\mathbf{0}) \right] \exp((L_\sigma B_{\mathbf{A}})^q L_x) \tag{149}$$

yields

$$\|z\|_{1\text{-var}} \leq C_1(\mathbf{G}_\psi) L_x. \tag{150}$$

This concludes the proof for $f_\theta$ with continuous input. For discretized inputs, the proof follows readily with constant $M_\Theta^D$.

## D.6 PROOF OF THEOREM B.5

For all $t \in [0, 1]$, we have the decomposition

$$w_t - v_t = w_0 - v_0 + \int_0^t \mathbf{F}(w_s)dx_s - \int_0^t \mathbf{G}(v_s)dr_s \tag{151}$$

$$= w_0 - v_0 + \int_0^t \big(\mathbf{F}(w_s) - \mathbf{F}(v_s)\big)dx_s + \int_0^t \mathbf{F}(v_s)dx_s - \int_0^t \mathbf{G}(v_s)dr_s \tag{152}$$

$$= w_0 - v_0 \tag{153}$$

$$+ \int_0^t \big(\mathbf{F}(w_s) - \mathbf{F}(v_s)\big)dx_s \tag{154}$$

$$+ \int_0^t \mathbf{F}(v_s)d(x_s - r_s) \tag{155}$$

$$+ \int_0^t \big(\mathbf{F}(v_s) - \mathbf{G}(v_s)\big)dr_s. \tag{156}$$

We control every one of these terms separately and conclude by applying Gronwall's Lemma. Writing

$$A := \left\| \int_0^t \big(\mathbf{F}(w_s) - \mathbf{F}(v_s)\big)dx_s \right\|,$$

$$B := \left\| \int_0^t \mathbf{F}(v_s)d(x_s - r_s) \right\|,$$

$$C := \left\| \int_0^t \big(\mathbf{F}(v_s) - \mathbf{G}(v_s)\big)dr_s \right\|$$

it is clear that

$$\|w_t - z_t\| \le \|w_0 - z_0\| + A + B + C. \tag{157}$$

**Control of the term $A$.** We have

$$A \le \int_0^t \|\mathbf{F}(w_s) - \mathbf{F}(v_s)\|_{\mathrm{op}} \|dx_s\|. \tag{158}$$

Since $F$ is $L_{\mathbf{F}}$-Lipschitz, this gives

$$A \le L_{\mathbf{F}} \int_0^t \|w_s - v_s\| \|dx_s\|. \tag{159}$$

**Control of the term $B$.** One gets using integration by parts

$$\int_0^t \mathbf{F}(v_s)d(x_s - r_s) = (x_t - r_t) - (x_0 - r_0) - \int_0^t (x_s - r_s)^\top d\mathbf{F}(v_s). \tag{160}$$

This gives

$$B \le \|x_0 - r_0\| + \|x - r\|_{\infty,[0,t]} \big(1 + \|\mathbf{F}(v)\|_{1\text{-var},[0,t]}\big). \tag{161}$$

Since $\mathbf{F}$ is $L_{\mathbf{F}}$-Lipschitz, one gets

$$\|\mathbf{F}(v)\|_{1\text{-var},[0,t]} \le L_{\mathbf{F}} \|v\|_{1\text{-var},[0,t]}. \tag{162}$$

Since $v$ is the solution to a CDE, we can resort to Proposition B.4 to bound its total variation. This yields

$$\|v\|_{1\text{-var},[0,t]} \le C_1(L_{\mathbf{F}}, \mathbf{F}, L_r) \|r\|_{1\text{-var},[0,t]} \tag{163}$$

and finally

$$B \le \|x_0 - r_0\| + \|x - r\|_{\infty,[0,t]} \big(1 + L_{\mathbf{F}}C_1(L_{\mathbf{F}}, \mathbf{F}, L_x) \|r\|_{1\text{-var},[0,t]}\big). \tag{164}$$

**Control of the term $C$.** Finally, we get that

$$C \le \int_0^t \|\mathbf{F}(v_s) - \mathbf{G}(v_s)\|_{\mathrm{op}} \|dr_s\| \tag{165}$$

$$\le \max_{v \in \Omega} \|\mathbf{F}(v) - \mathbf{G}(v)\|_{\mathrm{op}} L_r \tag{166}$$

where we recall that $\Omega$ is the closed ball

$$\Omega = \left\{ u \in \mathbb{R}^p \mid \|u\| \le (\|v_0\| + \|\mathbf{G}(0)\| L_r) \exp(L_{\mathbf{G}} L_r) \right\}.$$

Indeed, since $v$ is the solution of a CDE, its norm at any time $t \in [0, 1]$ is bounded as stated in Lemma 3.3. One can therefore bound the difference $\|\mathbf{F}(v_s) - \mathbf{G}(v_s)\|_{\mathrm{op}}$ by considering all possible values of $v$.

**Putting everything together.** Combining the obtained bounds on all terms, one is left with

$$\|w_t - v_t\| \le \|w_0 - v_0\| \tag{167}$$

$$+ \|x_0 - r_0\| + \|x - r\|_{\infty, [0,t]} \left( 1 + L_{\mathbf{F}} L_r C_1(L_{\mathbf{F}}, \mathbf{F}, L_x) \right) \tag{168}$$

$$+ L_r \max_{v \in \Omega} \|\mathbf{F}(v) - \mathbf{G}(v)\|_{\mathrm{op}} \tag{169}$$

$$+ L_{\mathbf{F}} \int_0^t \|w_s - v_s\| \|dx_s\|. \tag{170}$$

The proof is concluded by using Gronwall's Lemma A.4. If the path $(x_t)$ is continuous, we resort to its continuous version. If it is piecewise constant, we use its discrete version. The path $(\|w_t - v_t\|)_t$ only needs to be measurable and bounded. There are no assumptions on its continuity. This finally yields

$$\|w_t - v_t\| \le \Bigg( \|w_0 - v_0\| + \|x_0 - r_0\| \tag{171}$$

$$+ \|x - r\|_{\infty, [0,t]} \left( 1 + L_{\mathbf{F}} L_r C_1(L_{\mathbf{F}}, \mathbf{F}, L_x) \right) + \max_{v \in \Omega} \|\mathbf{F}(v) - \mathbf{G}(v)\|_{\mathrm{op}} L_r \Bigg) \tag{172}$$

$$\times \exp(L_{\mathbf{F}} L_x). \tag{173}$$

Now, notice that in our proof, the two CDEs are exchangeable. This means that we immediately get the alternative bound

$$\|w_t - v_t\| \le \Bigg( \|w_0 - v_0\| + \|x_0 - r_0\| \tag{174}$$

$$+ \|x - r\|_{\infty, [0,t]} \left( 1 + L_{\mathbf{G}} L_x C_1(L_{\mathbf{G}}, \mathbf{G}, L_r) \right) + \max_{v \in \Omega} \|\mathbf{F}(v) - \mathbf{G}(v)\|_{\mathrm{op}} L_x \Bigg) \tag{175}$$

$$\times \exp(L_{\mathbf{G}} L_r) \tag{176}$$

where we recall that

$$\Omega = \left\{ u \in \mathbb{R}^p \mid \|u\| \le (\|w_0\| + \|\mathbf{F}(0)\| L_x) \exp(L_{\mathbf{F}} L_x) \right\}.$$

### D.7 Proof of Theorem B.6

**Proof for continuous inputs.** We have

$$|f_{\theta_1}(x) - f_{\theta_2}(x)| = \left\| \Phi_1^\top z_1^1 - \Phi_2^\top z_1^2 \right\| \le \|\Phi_1 - \Phi_2\| \left\| z_1^1 \right\| + B_\Phi \left\| z_1^1 - z_1^2 \right\|. \tag{177}$$

Using Theorem B.5, one gets that

$$\left\| z_1^1 - z_1^2 \right\| \le \exp\left( (L_\sigma B_{\mathbf{A}})^q L_x \right) \left( \left\| z_0^1 - z_0^2 \right\| + L_x \left\| \mathbf{G}_{\psi^1} - \mathbf{G}_{\psi^2} \right\|_{\infty, \Omega} \right), \tag{178}$$

where

$$\Omega = \left\{ u \in \mathbb{R}^p \mid \|u\| \le \left( L_\sigma (B_{\mathbf{U}} B_x + B_v) + \|\mathbf{G}_\psi(\mathbf{0})\|_{\text{op}} \right) \exp \left( (L_\sigma B_{\mathbf{A}})^q L_x \right) \right\}.$$

The quantity $\|\mathbf{G}_\psi(\mathbf{0})\|$ is bounded by Lemma B.2. The radius of the ball $\Omega$ is thus bounded by a constant that does not depend on the value of the network's parameters, but only on the constants defining $\Theta$.

We first bound

$$\left\| z_0^1 - z_0^2 \right\| \le L_\sigma \|\mathbf{U}_1 - \mathbf{U}_2\| B_x + L_\sigma \|v_1 - v_2\|. \tag{179}$$

One also has that

$$\left\| \mathbf{G}_{\psi^1} - \mathbf{G}_{\psi^2} \right\|_{\infty, \Omega} \le \max_{z \in \Omega} \sum_{j=1}^{q} \left[ C_{\mathbf{b}}^j \left\| \mathbf{b}_j^1 - \mathbf{b}_j^2 \right\| + C_{\mathbf{A}}^j(z) \left\| \mathbf{A}_j^1 - \mathbf{A}_j^2 \right\| \right]. \tag{180}$$

Since $\Omega$ is bounded, $(C_{\mathbf{b}}^j)$ is bounded and the functions $(C_{\mathbf{A}}^j(z))$ are continuous with respect to $z$, they are bounded on $\Omega$. Taking maximas, we get

$$\left\| \mathbf{G}_{\psi^1} - \mathbf{G}_{\psi^2} \right\|_{\infty, \Omega} \le \max_{1 \le i \le q} C_{\mathbf{b}}^i \sum_{j=1}^{q} \left\| \mathbf{b}_j^1 - \mathbf{b}_j^2 \right\| + \max_{z \in \Omega, 1 \le i \le q} C_{\mathbf{A}}^i \sum_{j=1}^{q} \left\| \mathbf{A}_j^1 - \mathbf{A}_j^2 \right\|. \tag{181}$$

Since $z_1^1$ is bounded as the endpoint of a NCDE, one gets using Theorem B.5 that

$$\|\Phi_1 - \Phi_2\| \left\| z_1^1 \right\| \le B_\Phi L_\sigma \exp \left( (B_{\mathbf{A}} L_\sigma)^q L_x \right) \left( B_{\mathbf{U}} B_x + B_v + \kappa_\Theta(\mathbf{0}) L_x \right) \|\Phi_1 - \Phi_2\| \tag{182}$$

Putting everything together, one gets that

$$\|f_{\theta_1}(x) - f_{\theta_2}(x)\| \le M_\Theta \|\Phi_1 - \Phi_2\| \tag{183}$$

$$+ C_{\mathbf{A}} \sum_{j=1}^{q} \left\| \mathbf{A}_j^1 - \mathbf{A}_j^2 \right\| + C_{\mathbf{b}} \sum_{j=1}^{q} \left\| \mathbf{b}_j^1 - \mathbf{b}_j^2 \right\| + \tag{184}$$

$$+ C_{\mathbf{U}} \|\mathbf{U}_1 - \mathbf{U}_2\| + C_v \|v_1 - v_2\| \tag{185}$$

with

$$M_\Theta := B_\Phi L_\sigma \exp \left( (B_{\mathbf{A}} L_\sigma)^q L_x \right) \left( B_{\mathbf{U}} B_x + B_v + \kappa_\Theta(\mathbf{0}) L_x \right), \tag{186}$$

$$C_{\mathbf{A}} := B_\Phi L_x \exp \left( (L_\sigma B_{\mathbf{A}})^q L_x \right) \times \max_{z \in \Omega, 1 \le i \le q} C_{\mathbf{A}}^i(z) \tag{187}$$

$$C_{\mathbf{b}} := B_\Phi L_x \exp \left( (L_\sigma B_{\mathbf{A}})^q L_x \right) \times \max_{1 \le i \le q} C_{\mathbf{b}}^i \tag{188}$$

$$C_{\mathbf{U}} := B_\Phi B_x \exp(L_\sigma B_{\mathbf{A}} L_x) L_\sigma \tag{189}$$

$$C_v := B_\Phi \exp(L_\sigma B_{\mathbf{A}} L_x) L_\sigma. \tag{190}$$

which concludes our proof.

**Proof for discretized inputs.** The proof for $|f_{\theta_1}(\mathbf{x}^D) - f_{\theta_2}(\mathbf{x}^D)|$ follows in a similar fashion, but with discretization dependant constants. Indeed,

$$|f_{\theta_1}(\mathbf{x}^D) - f_{\theta_2}(\mathbf{x}^D)| \le \left\| \Phi_1^\top z_1^1 - \Phi_2^\top z_1^2 \right\| \le \|\Phi_1 - \Phi_2\| \left\| z_1^{D,1} \right\| + B_\Phi \left\| z_1^{D,1} - z_1^{D,2} \right\|. \tag{191}$$

By Theorem 3.3, we have

$$\left\| z_1^{D,1} \right\| \le L_\sigma \left( L_\sigma B_{\mathbf{U}} B_x + B_v + L_\sigma B_{\mathbf{b}} \frac{(L_\sigma B_{\mathbf{A}})^q - 1}{L_\sigma B_{\mathbf{A}}} L_x \right) \left( 1 + (L_\sigma B_{\mathbf{A}})^q L_x |D| \right)^K. \tag{192}$$

To bound the difference $\left\| z_1^{D,1} - z_1^{D,2} \right\|$, we resort to Theorem B.5 which gives us

$$\left\| z_1^{D,1} - z_1^{D,2} \right\| \leq \exp\left( (L_\sigma B_{\mathbf{A}})^q L_x \right) \left( \left\| z_0^1 - z_0^2 \right\| + L_x \left\| \mathbf{G}_{\psi^1} - \mathbf{G}_{\psi^2} \right\|_{\infty,\Omega^D} \right). \tag{193}$$

Remark that the ball $\Omega^D$ depends on the discretization. Indeed, the diameter of this set upper bounds the value of $z_1^D$ for all possible controlled ResNets in $\Theta$ and initial values of $\mathbf{x}^D$, which depend on the discretization. As for continuous inputs, we have

$$\left\| z_0^1 - z_0^2 \right\| \leq L_\sigma \left\| \mathbf{U}_1 - \mathbf{U}_2 \right\| B_x + L_\sigma \left\| v_1 - v_2 \right\|. \tag{194}$$

Now, one also has that

$$\left\| \mathbf{G}_{\psi^1} - \mathbf{G}_{\psi^2} \right\|_{\infty,\Omega^D} \leq \max_{z \in \Omega^D} \sum_{j=1}^{q} \left[ C_{\mathbf{b}}^j \left\| \mathbf{b}_j^1 - \mathbf{b}_j^2 \right\| + C_{\mathbf{A}}^j(z) \left\| \mathbf{A}_j^1 - \mathbf{A}_j^2 \right\| \right] \tag{195}$$

and taking maximas again but this time on $\Omega^D$, one is left with

$$\| f_{\theta_1}(x) - f_{\theta_2}(x) \| \leq M_\Theta^D \left\| \Phi_1 - \Phi_2 \right\| \tag{196}$$

$$+ C_{\mathbf{A}}^D \sum_{j=1}^{q} \left\| \mathbf{A}_j^1 - \mathbf{A}_j^2 \right\| + C_{\mathbf{b}}^D \sum_{j=1}^{q} \left\| \mathbf{b}_j^1 - \mathbf{b}_j^2 \right\| + \tag{197}$$

$$+ C_{\mathbf{U}} \left\| \mathbf{U}_1 - \mathbf{U}_2 \right\| + C_v \left\| v_1 - v_2 \right\| \tag{198}$$

with constants

$$M_\Theta^D := L_\sigma \left( L_\sigma B_{\mathbf{U}} B_x + B_v + L_\sigma B_{\mathbf{b}} \frac{(L_\sigma B_{\mathbf{A}})^q - 1}{L_\sigma B_{\mathbf{A}}} L_x \right) \left( 1 + (L_\sigma B_{\mathbf{A}})^q L_x |D| \right)^K \tag{199}$$

$$C_{\mathbf{A}}^D := B_\Phi L_x \exp\left( (L_\sigma B_{\mathbf{A}})^q L_x \right) \times \max_{z \in \Omega^D, 1 \leq i \leq q} C_{\mathbf{A}}^i(z) \tag{200}$$

$$C_{\mathbf{b}}^D := B_\Phi L_x \exp\left( (L_\sigma B_{\mathbf{A}})^q L_x \right) \times \max_{1 \leq i \leq q} C_{\mathbf{b}}^i \tag{201}$$

$$C_{\mathbf{U}} := B_\Phi B_x \exp(L_\sigma B_{\mathbf{A}} L_x) L_\sigma \tag{202}$$

$$C_v := B_\Phi \exp(L_\sigma B_{\mathbf{A}} L_x) L_\sigma. \tag{203}$$

# E   EXPERIMENTAL DETAILS

## E.1   EXPERIMENT 1: ANALYSING THE TRAINING DYNAMICS

**Time series.** The paths are 4-dimensional sample paths from a fBM with Hurst parameter $H = 0.7$, sampled at 100 equidistant time points in the interval $[0, 1]$. The paths are generated using the Python package `stochastic`. We include time as a supplementary channel, as advised in Kidger et al. (2020).

**Well-specified model.** The output is generated from a shallow teacher NCDE

$$dz_t^\star = \text{Tanh}\Big(\mathbf{A}z_t^\star + \mathbf{b}\Big)dx_t$$

initialized with

$$z_0^\star = \mathbf{U}x_0 + v$$

with random parameters for which $p = 3$.

**Model.** We train a student NCDE with identical parametrization. The model is initialized with Pytorch's default initialization. In the **first and second figures** (starting from the left), the model is trained for 2000 iterations with Adam. We use the default values for $\alpha, \beta$ and a learning rate of $5 \times 10^{-3}$. The size of the training sample is set to $n = 100$. For the **third figure**, we run 25 training runs on the same dataset, for the same model, with 1000 training iterations and record the normalized value of the parameters at the end of training.

## E.2   EXPERIMENT 2: EFFECT OF THE DISCRETIZATION

**Time series.** We consider a classification task, in which a shallow NCDE is trained to distinguish between two dimensional fBMs with respective Hurst parameters $H = 0.4$ ($y = 0$) and $H = 0.6$ ($y = 1$). Time is included as a supplementary channel.

**Model.** For the **first figure**, we train a shallow NCDE classifier with $p = 3$ on $n = 100$ time series sampled at 100 equidistant time points in $[0, 1]$ for 100 iterations with Binary Cross Entropy (BCE) loss. We use Adam with default settings and a learning rate of $5 \times 10^{-2}$. For the **second and third** figures, we fix the initialization and the last linear layer $\Phi$ of the NCDE and perform 300 training runs with Adam (default parameters and learning rate of $1 \times 10^{-4}$). For each run, we randomly downsample $n = 50$ time series by selecting $K = 5$ sampling points, which always include 0 and 1, among the 200 initial points. The model is then trained for 600 iterations, and the expected generalization error is computed on 50 test samples. We freeze the initialisation layer and $\Phi$ following Marion (2023) to isolate the effect of the discretization on the generalization error.

# F    SUPPLEMENTARY ELEMENTS

## F.1    PRACTICAL AND THEORETICAL IMPLICATIONS OF OUR RESULTS

We see three main practical and theoretical implications of our results.

**Designing new architectures for NCDEs.**    Our paper offers performance guarantees and sheds light on the theoretical behaviour of NCDEs. We think that a promising application of our work lies in the design of new architectures for the neural vector field $\mathbf{G}_\psi$. In our work, as in most applications of NCDEs, this neural vector field is chosen as a simple MLP. However, a recent and rich literature has highlighted the benefits of more structured architectures (for instance through orthogonal weights - see Li et al. (2019)). Our work could help to understand the theoretical implications of such extensions to NCDEs. On a sidenote, we would like to mention the recent article by Chiu et al. (2023), which was unpublished by submission date, and which takes a first step in this direction by using a U-Net as a neural vector field for learning from video data.

**Quantifying the impact of sparse sampling.**    On an applied level, our results can serve as a guideline to practitioners that use irregular time series. For instance, from our personal experience, MDs often wonder whether they have collected enough longitudinal data in a clinical setting to conduct inference with NCDEs. Our results on the sampling bias may help to make an informed decision in such a case. Additionally, our results can help a practitioner facing a "decide now or sample more" dilemma, in which an agent needs to decide whether she defers a decision in order to acquire more information through time-consuming sampling. See for instance Schäfer & Leser (2020) for a discussion of similar problems.

**Connexion to Control Theory.**    Control Theory can be seen as an inversion of the setup of NCDEs as noted by Kidger (2022) (see Section 3.1.6.2). In control theory, the dynamics are fixed (known or unknown) and one tries to optimize the input path. In our setup, the path is fixed and the dynamics are optimized. We think, however, that our results can transfer to control theory: our theoretical framework can indeed accommodate many situations in which one learns with continuous time data. Consider, for instance, a setup with fixed dynamics in which one minimizes a criterion by optimizing over the space of paths by choosing the vector fields of a neural SDE — see for instance Zhang et al. (2022) for a close setup. One could use our results on the continuity of the flow to bound the difference between two outputs - driven by two paths with different vector fields - by the difference between the two vector fields, and hence derive an upper bound on the covering number of such a class of predictors. To perform this extension, one would however need to leverage additional tools from stochastic calculus and rough analysis.

## F.2    COMPARISON WITH EARLIER RESULTS BY CHEN ET AL. (2020A)

Questions about the comparison of our results with earlier results by Chen et al. (2020a) were raised during the reviewing process of this article. We provide a thorough discussion of these questions in this section. The notations used are the ones by the authors of the aforementioned article and we refer to the AISTATS 2020 version of their work.
In their work, Chen et al. (2020a) consider so-called vanilla RNN which embed for every individual $i = 1, \ldots, n$ a time series $(x_{i,t})_{t=1}^T$ through

$$h_{i,t} = \sigma_h\big(U h_{i,t-1} + W x_{i,t}\big). \tag{204}$$

Here, $U \in \mathbb{R}^{d_h \times d_h}$ and $W \in \mathbb{R}^{d_h \times d_x}$ are learnable parameters. This latent state is then in turn used to produce a prediction about the true label $(z_i, t)_{t=1}^T$ by computing

$$y_{i,t} = \sigma_y\big(V h_{i,t}\big) \tag{205}$$

where $V \in \mathbb{R}^{d_z \times d_h}$ is again a learnable parameter. In their work, the authors derive generalization bounds by leveraging similar techniques to ours. Indeed, they first proceed to show that vanilla RNN are Lipschitz with respect to their parameters (see their Lemma 2, in Section 3 page 4 left column). This result is similar to our Theorem B.6. They then proceed to derive the covering number of vanilla RNN (see their Lemma 3, Section 3, page 5 left column). This is identical to our Proposition 4.2.

Finally, as in our article, Chen et al. are able to derive the Rademacher complexity using Dudley's entropy integral.

Let us now discuss the generalization bound in our two works. The first kind of generalization bound of Chen et al. (2020a) scales with the number of parameters in their model equal to

$$d = \sqrt{d_x d_h + d_h^2 + d_h d_y} \tag{206}$$

which is expected when using Lipschitz based complexity measures (see their Theorem 2). This is similar to our results in Theorem 4.1, where the complexity term also scales with the number of parameters.

In a second time, Chen et al. (2020a) prove more refined generalization bounds by controlling the $(2, 1)$ norm of the weight matrices (see their Section 4). They obtain two Theorems, yielding a scaling either in

$$\sqrt{d} \log \sqrt{d} \tag{207}$$

or

$$\sqrt{d \log d} \tag{208}$$

using again the original notations. This bounds are stronger than the previous ones, and than our bounds, since their involve the $4$-th root of the number of parameters. They do, however, require stronger assumptions on the norm of the parameters. We leave an application of this kind of results to our setup to future work.

### F.3  Comparaison with earlier results by Golowich et al. (2018)

Golowich et al. (2018) deploy new techniques, based on a refined control of the Schatten norms of the weight matrices, that allows to obtain depth-independent bounds. The authors consider deep non-residual neural networks of the form

$$x \mapsto \sigma \circ \mathbf{W}_L \circ \sigma \circ \mathbf{W}_{L-1} \circ \cdots \circ \sigma \circ \mathbf{W}_1(x) \tag{209}$$

where $\left(\mathbf{W}_k\right)_{k=1}^{L}$ are learnable parameters. The bounds they obtain scale as

$$\mathcal{O}\left(\frac{\Pi_F \sqrt{\log\left(\Pi_F / \pi_S\right)}}{n^{1/4}}\right) \tag{210}$$

where $\Pi_F$ is an upper bound on the product of Froebenius norms of the weight matrices and $\pi_S$ is a lower bound on the product of the spectral norms of the weight matrices. To obtain depth-independent bound, the authors require these two quantities to be depth independent. Interestingly, observe that these generalization bounds come at the price of a worsened sample complexity (in the sense of dependence on the number of samples). The interested reader may also consider the discussion of this point provided in Marion (2023).

To discuss these bounds and to avoid any confusion, we first need to make a clear distinction between the **depth** and the **length** of our NCDE predictor:

- The **length** of our controlled ResNet (or NCDE with piecewise constant path embedding) refers to the number of sampling times and is equal to $K$. In ResNets (see Appendix F.4), this quantity is usually referred to as **depth**, hence a possible confusion. Contrarily to ResNets, this parameter is data dependant i.e. it depends on the number of sampling times of the time series at hand.

- The **depth** of our NCDE refers to the depth of the neural vector field $\mathbf{G}_\psi$, which is assumed in our work to be a standard MLP.

Our bounds are not **depth** independent since increasing the depth of the vector field $\mathbf{G}_\psi$ worsen our generalization bounds which scale with the square root of the number of parameters.

We conjecture that one could apply the bounds obtained by Golowich et al. (2018) to provide a refined generalization bound on the complexity stemming from the neural vector field $\mathbf{G}_\psi$. However, we believe that since in practice NCDEs are used with neural vector fields for which $q \leq 3$, trading depth-independence for a worsened convergence speed in the number of samples will most likely not benefice our bounds.

F.4   COMPARAISON OF RESNETS AND NCDES

A vanilla ResNet architecture can be described as an embedding of $x \in \mathbb{R}^d$ through

$$h_0 = \mathbf{U}x \in \mathbb{R}^p \tag{211}$$

$$h_{k+1} = h_k + \frac{1}{L^\beta}\mathbf{W}_k\mathbf{G}_\psi(h_k) \text{ for all } k = 0, \dots, L \tag{212}$$

$$\hat{y} = \Phi^\top h_L \tag{213}$$

where $U \in \mathbb{R}^{d\times p}$, $(\mathbf{W}_k)_{k=0}^{L-1} \in \mathbb{R}^{p\times p}$, $\mathbf{G}_\psi : \mathbb{R}^p \to \mathbb{R}^p$ is a neural network parametrized by $\psi$ and $\hat{y}$ is the final prediction of the ResNet. The scaling factor $\beta$ is usually chosen between $1/2$ and $1$.

Consider now the *controlled* ResNet architecture, which can be described as mentioned earlier as an embedding of a time series $\mathbf{x} = (\mathbf{x}_0, \dots, \mathbf{x}_{t_K}) \in \mathbb{R}^{d\times(K+1)}$ through

$$z_0 = \mathbf{U}x_0 \in \mathbb{R}^p \tag{214}$$

$$z_{k+1} = z_k + \mathbf{G}_\psi(z_k)\Delta\mathbf{x}_{t_{k+1}}^D \text{ for all } k = 0, \dots, K-1 \tag{215}$$

$$\hat{y} = \Phi^\top z_K \tag{216}$$

where we have simplified, for the sake of the analogy, the initialization to be a linear initialization layer.

One can see that, from an architectural point of view, the layer-dependant weights $(\mathbf{W}_k)_{k=0}^{L-1}$ of the ResNet play a similar role to the increments of the time series $(\Delta\mathbf{x}_{t_j})_{j=1}^K$. However, in ResNets these weights are learned, whereas in *controlled* ResNets and NCDEs they are given through the embedded time series. Also remark that while the weights of a ResNet are shared between all inputs (i.e. two inputs $x$ and $x'$ are embedded using the same weights), the increments of time series are, by definition, individual specific. This allows NCDEs to have individual-specific dynamics for the latent state.

This allows us to compare our experimental findings with the ones of Marion (2023), who observes that the generalization error correlates with the Lipschitz constant of the weights defined as

$$\max_{0\le k\le L_1} \|\mathbf{W}_{k+1} - \mathbf{W}_k\|.$$

In our setting, this translates to the fact that the generalization error is correlated with the average absolute path variation

$$n^{-1}\sum_{i=1}^n \max_{k=1,\dots,K-1} \left\|\mathbf{x}_{t_{k+1}}^{D,i} - \mathbf{x}_{t_k}^{D,i}\right\| = n^{-1}\sum_{i=1}^n \max_{k=1,\dots,K} \left\|\Delta\mathbf{x}_{t_k}^D\right\|.$$

Remark that we have to consider the *average* over individuals, since the increments are individual specific. Since these increments are not trained, and since we make the assumption that they are bounded by $L_x|D|$, we should also observe a correlation with $L_x|D|$ - see Figure 4. Our code is available at this link.

