# OpenReview forum: "On the Generalization and Approximation Capacities of Neural Controlled Differential Equations"
_ICLR.cc/2024/Conference — ICLR 2024 poster_

### Official Review · Reviewer_V1D4 · 2023-10-29

**Soundness:** 3 good
**Presentation:** 2 fair
**Contribution:** 3 good
**Rating:** 6
**Confidence:** 3

**Summary:**

This paper studies the generalization bound for neural-controlled differential equations(NCDE). Given an NCDE that minimizes the empirical risk from some sampled data, the author first derives a generalization bound between the empirical and expected risk given the sampling process for the input time series, and then an upper bound on the difference between the expected risk and the optimal risk from the ground truth predictor.

**Strengths:**

1. novel generalization bound for neural-controlled differential equations.
2. an upper bound on the excessive risk from sampling-induced bias and approximation bias.

**Weaknesses:**

1. The presentation in Theorem 4.1 could be improved, as it is hard to see how the bound explicitly depends on depth $q$, and discretization $D$. Right now it is hidden in the constant $C_q$ and $K_1^D, K_2^D$.
2. The experiments seem very preliminary: the sampled time series only have $K=5$ points. I am not sure how interesting it is to investigate this regime as the generalization error would be poor with this few samples.

**Questions:**

1. Any assumption on the continuous $x$, or its distribution, if it is random.
2. $|D|$ appears in Remark 3.5, but defined much later.

---

> ### Author Response · Authors · 2023-11-16
> **Answer to Reviewer V1D4**
>
> We thank the reviewer for acknowledging our contributions and results. Let us first address the two mentioned weaknesses. We agree that the presentation of Theorem 4.1 can be improved and we will try to do so within the page constraints. In particular, as this has also been highlighted by Reviewer 1, we will make the constants explicit. Concerning the experimental part, we make two remarks. First, as the discretization gap $|D|$ vanishes at linear speed - i.e. $1/M$ when sampling $M$ points in a equidistant fashion - the effect of discretization is almost null when short time series are sampled at high frequency. We thus have chosen this low sampling regime to obtain a visible effect. Secondly, our article is of theoretical nature ; we have chosen to give an extensive presentation of our theoretical results, which are complex and require long developments, instead of a more developed experimental section. We are planning on conducting a follow-up work benchmarking various methods designed for supervised learning with irregular time series, which will be the empirical companion paper to this work. We also refer the reader to the growing body of applied works on NCDEs, which we extensively cite in the introduction.
>
> We now answer the two questions raised by the reviewer. First, the underlying continuous path $x$ is supposed to be unknown and observed at a finite number of sampling times. It is continuous of bounded variation, which is required for NCDEs to be well defined (Assumptions 1 and 2). We make the simplifying assumption that the paths are Lipschitz, but this can be weakened to include paths paths of bounded variation with non linear modulus of continuity. We work in a fixed design setting, i.e. we consider our data to be fixed throughout the article. As highlighted in our response to Reviewer 2, this stands in contrast to settings in stochastic control theory, where one learns over the distribution of random paths (i.e. stochastic processes with parametrized vector fields). Secondly, we thank the reviewer for catching the typo on $|D|$, which we have fixed.

---

### Official Review · Reviewer_QrUR · 2023-10-30

**Soundness:** 3 good
**Presentation:** 3 good
**Contribution:** 3 good
**Rating:** 8
**Confidence:** 4

**Summary:**

The manuscript presents a theoretical investigation into the generalization and approximation capacities of neural controlled differential equations (NCDEs). It aims to answer critical questions about the generalization and approximation capabilities of NCDEs and how these are affected by irregular sampling. The theoretical contributions are backed by experiments. The authors provide a rigorous mathematical framework to offer insights into the capabilities and limitations of NCDEs.

**Strengths:**

Originality:
The paper stands out for its focus on providing a theoretical framework for NCDEs. This can serve as a foundational work for further investigations.
Quality:
The mathematical rigor of the paper is high, with well-laid-out proofs and theoretical discussions.
Clarity:
Despite the mathematical complexity, the paper is organized in a way that progressively builds up the reader's understanding of the topic.
Significance:
The theoretical insights offered could be highly impactful, providing guidelines for both practical applications and future research in machine learning and control theory.

**Weaknesses:**

In Abstract: ``However, no theoretical analysis of their performance has been provided yet''. This is not true.

The paper could do a better job of situating its contributions within the existing literature, especially to clarify its novelty.

While the focus is theoretical, some discussion on the practical implications of these theories could make the paper more well-rounded.

There are some grammar/format issues, such as, ``since y is the sum of a linear transformation of the endpoint of a CDE an a noise term ε bounded by …’’

**Questions:**

Could you clarify how your theoretical contributions on the approximation part diverge from existing works on the approximation ability of NCDE like [1] or [2]?

[1] Patrick Kidger. On neural differential equations. arXiv preprint arXiv:2202.02435, 2022.

[2] Patrick Kidger, James Morrill, James Foster, and Terry Lyons. Neural controlled differential equations for irregular time series. Advances in Neural Information Processing Systems, 33:6696–6707, 2020.

What are the potential practical applications that could benefit from the theoretical insights provided in this paper? Could these insights lead to more efficient or accurate NCDE models?

---

> ### Author Response · Authors · 2023-11-16
> **Answer to Reviewer QrUR**
>
> We warmly thank the reviewer for acknowledging the originality, quality and clarity of our work.
>
> **Relation to earlier work.** We agree that our presentation of the related work by Kidger (2022) and Kidger (2020) is incomplete and that the abstract is misleading. The contributions of these two works can be summarized as follows. First, the authors proves the existence of solutions to neural CDEs. This result follows from the Picard-Lindelhöf theorem. Secondly, the authors of this article also prove a universal approximation property for neural CDEs i.e. that sufficiently wide NCDEs can approximate any continuous function mapping the space of paths to $\mathbb{R}$ (Theorem 3.1. in Kidger 2020, and Theorem C.25 in Kidger 2022). We have emphasised these contributions in the abstract and the introduction. In contrast to this, our approximation bound provides a precise decomposition of the approximation error in the setup where the function linking the path to the outcome is a CDE.
>
> We also thank the reviewer for noticing typos, which we have fixed.
>
> **Implications of our work.** A deeper discussion of practical implications is indeed needed, as suggested by the reviewer. We see three principal implications of our work.
>
> 1. **Designing new architectures for NCDEs.** Our paper offers performance guarantees and sheds light on the theoretical behaviour of NCDEs. We think that a promising application of our work lies in the design of new architectures for the neural vector field $\mathbf{G}_\psi$. In our work, as in most applications of NCDEs, this neural vector field is chosen as a simple MLP. However, a recent and rich literature has highlighted the benefits of more structured architectures (for instance through orthogonal weights - see Jia 2019). Our work could help to understand the theoretical implications of such extensions to NCDEs. On a sidenote, we would like to mention the recent article by Chiu et al. ("Exploiting Inductive Biases in Video Modeling through Neural CDEs", 2023), which was unpublished by submission date, and which takes a first step in this direction by using a U-Net as a neural vector field for learning from video data.
>
> 2. **Quantifying the impact of sparse sampling.** On an applied level, our results can serve as a guideline to practitioners that use irregular time series. For instance, from our personal experience, MDs often wonder whether they have collected enough longitudinal data in a clinical setting to conduct inference with NCDEs. Our results on the sampling bias may help to make an informed decision in such a case. Additionally, our results can help a practitioner facing a "decide now or sample more" dilemma, in which an agent needs to decide whether she defers a decision in order to acquire more information through time-consuming sampling (see for instance Schäfer, "TEASER: early and accurate time series classification", 2020, for a discussion of similar problems).
>
> 3. **Links with Control Theory.** The reviewer mentions control theory, which can be seen as an inversion of the setup of NCDEs as noted by Kidger 2022 (see Section 3.1.6.2). In control theory, the dynamics are fixed (known or unknown) and one tries to optimise the input path. In our setup, the path is fixed and the dynamics are optimised. We think, however, that our results can transfer to control theory: our theoretical framework can indeed accommodate many situations in which one learns with continuous time data. Consider, for instance, a setup with fixed dynamics in which one minimises a criterion by optimising over the space of paths by choosing the vector fields of a neural SDE (see Zhang, "Neural Stochastic Control", 2022 for a close setup). One could use our results on the continuity of the flow to bound the difference between two outputs - driven by two paths with different vector fields - by the difference between the two vector fields, and hence derive an upper bound on the covering number of such a class of predictors. To perform this extension, one would however need to leverage additional tools from stochastic calculus and rough analysis.
>
> To satisfy page constraints,**we have included these remarks in Appendix F.1** and include a reference to this part in the conclusion. If the reviewer sees other implications, we would be more than happy to include them as well.

---

> > ### Comment · Reviewer_QrUR · 2023-11-21
> >
> > Thanks for your reply.

---

### Official Review · Reviewer_c9Ry · 2023-10-31

**Soundness:** 3 good
**Presentation:** 3 good
**Contribution:** 2 fair
**Rating:** 5
**Confidence:** 3

**Summary:**

This paper makes a first-of-its-kind attempt at bounding the generalization error of NCDE.

**Strengths:**

The analysis done is thorough and it is definitely an achievement to have setup the formalism required to compute a generalization bound like this.

**Weaknesses:**

In Section 4, Theorem 4.1 the upper bound for the generalization error varies with both the width and depth of the neural network and unlike Chen et al. (2019a), where the growth w.r.t the depth and width is logarithmic, in this case it’s polynomial. This exponential blow-up in the value of bound makes the whole exercise somewhat underwhelming.

Also its worth mentioning that the discretization dependence in Theorem 4.1 is through the constants K1^D and K2^D
 - and these two critical quantities don't seem defined anywhere in the main paper! This makes the theorem highly opaque!

The entire point why a generalization error bound is interesting is because it reveals the generalization error to be independent of some natural size parameter of the system. Like, most often the baseline achievement of a Rademacher bound for a neural system is to show that at a fixed depth and norm bound on the weights involved the generalization is width independent. In this bound, in this work, there is absolutely no such thing happening - the bound in Theorem 4.1 is worsening with every such parameter with which naively one would have expected worsening to happen!

The numerical Illustrations also seem incomplete in the absence of any demonstration of how well the bounds hold up in the cases mentioned there. In the bare minimum, in Figure 4 kind of experiments one would have expected to see a scatter plot of the theoretical bound and the true generalization error as the sampling gap, time and avg. max path variation is changed. Such a scatter plot would have revealed if there is at all any correlation between the theoretical bound and the truth.

**Questions:**

Q1.
Can the authors explain why there is an exponential degradation of the bound's dependence on width as compared to the result in Chen et al. (2019a), which seems to be the closest literature to this? Can you argue that this degradation is inevitable?

Q2.
Can the authors point out as to what is the surprise or the non-triviality about the dependencies that are visible in the bound given in Theorem 4.1?

---

> ### Author Response · Authors · 2023-11-16
> **Answer to Reviewer c9Ry**
>
> We thank the reviewer for this thorough review of our paper and its theoretical part which raises interesting questions, and for acknowledging our contributions. Before turning to the core of the review, we agree that giving the explicit form of all constants will make Theorem 4.1. more clear, as it has also been noted by Reviewer 3. We have added references to the constants in Theorem 4.1.
>
> **Comparison with earlier work by Chen et al. (2020).** We first address the comparison made with the earlier results of Chen et al. ("On Generalization Bounds of a Family of Recurrent Neural Networks", 2020 - we refer to AISTATS version from now on). We would first like to emphasise that the parameter $q$, which in our setting corresponds to the depth of the neural vector field applied at each update of the latent state, is completely absent from the aforementioned article: Chen et al. only consider shallow vector fields (i.e. "vanilla RNN" in their phrasing). Their neural vector field has no hidden layers but is only parameterized by matrices $W$ and $U$ which have resp. $d_h \times d_x$ and $d_h \times d_h$ parameters. We strongly encourage the reviewer to compare the definition of Chen et al.'s vector field (p.2, left column) with ours (Equation 1, page 4) to convince himself of the difference between our models.
>
> With this clarification in mind, we argue that the first bounds obtained by Chen et al. and ours are of identical nature. Let us take a close look at Theorem 2 in Chen et al. (p. 4, left column): one can see that the complexity term scales as the square root of the number of parameters $d$, times some log factors. We obtain a similar result (see our Theorem 4.1 on page 6): the complexity term, in our bound, grows as the square root of the number of parameters times some log factors. As stressed above, the factor $q$ appears here since having $q$ layers in the neural vector field multiplies the number of parameters by $q$. This is not a surprise since both works use the same fundamental technique (Lipschitz-based complexity measures). In the last part of their paper, Chen et al. provide a slightly improved bound at the price of stronger restrictions on the parameter space. We leave an extension of these techniques to NCDEs to future work. We thank the reviewer for noticing this precise point, which we had missed. **We have added a mathematical and precise comparison of the results in Appendix F.2**.

---

> > ### Author Response · Authors · 2023-11-16
> > **Answer to Reviewer c9Ry**
> >
> > **On generalisation.** We now wish to precise what we mean by generalisation, which might differ from the reviewer's definition. In our understanding, a generalisation bound simply ensures convergence of the empirical loss towards the expected loss. It does not need to be independent of some parameters of the system, even though one can obtain such dimension-free bounds in special setups (f.i. when learning with linear models) and with certain special techniques. For instance, if one considers vanilla bounds in linear regression with deterministic non-linear transformations of the input data (see Bach, "Learning Theory from First Principles", 2021, Chapter 4 Section 5) there is no independence from the size of the parameter space: the complexity of the predictors scales with the square root of the log of the number of features.
> >
> > However, we are aware that some other types of generalisation bounds exist for deep neural networks that may achieve bounds with very weak or no dependence on some of the parameters of the network. Some of these techniques rely on a close analysis of the optimisation process and the implicit bias of optimizers, which is not the objective of our paper. See for instance Gunasekar et al. ("Implicit Regularization in Matrix Factorization", 2017) or Soudry et al. ("The Implicit Bias of Gradient Descent on Separable Data", 2017) for an example of such an analysis. Another type of bound pioneered by Golovich et al. (2018) allows indeed to obtain depth independent bounds at the price of worsened sample complexity: convergence of the empirical loss towards the expected loss is ensured at speed $n^{1/4}$ instead of the more common speed $n^{1/2}$. We believe that one could apply these bounds here to obtain a refined analysis of the complexity induced by the depth of the neural vector field at the price of degraded sample complexity. However, since NCDEs are used with very shallow neural networks i.e. $q\leq 3$, we believe that this trade-off will not improve our bounds. **We have provided a mathematical description of these points in Appendix F.3.** and leave a deeper analysis of this problem for future work.
> >
> > **On experiments.** Concerning our experiments, it is unclear to us what the reviewer means by the theoretical value of the bounds. Many of the quantities provided in our bounds are unknown (i.e. data distribution dependant) constants and cannot easily be estimated. Let us emphasise again what our results show. First, experiment 1 shows that NCDEs can achieve zero loss (i.e. perfect interpolation and generalisation) with bounded parameters. This implies that the crucial assumption of bounded parameter space is satisfied in practice, and that the bounds lie in a reasonable range. A negative result would have been that NCDEs need weights with extremely high magnitude to achieve good performance. Our second result shows that the generalisation gap correlates with both the discretization gap and with a quantity that we have called the average maximum path variation.
> >
> > We hope that these explanations make our contribution more clear. We would like to thank the reviewer for raising questions on these points, which lead us to add thorough discussions of the results mentioned above to our article's Appendix. We will be happy to provide more explanation and supplementary details if needed.

---

> ### Comment · Reviewer_c9Ry · 2023-11-18
> **Thanks!**
>
> Let me explain again what the issue is with the experiments :
> the second and the third figures of Figure 4 are the only place where the true generalization error is plotted - but we have no idea from this plot what the theoretical bound is for these same experiments. The empirical Rademacher complexity should have been computable on the trained nets considered in these experiments.
>
> The authors have themselves said the following in the added response at the bottom of page 36,
> "Our bounds are not depth independent since increasing the depth of the vector field Gψ worsen our generalization bounds which scale with the square root of the number of parameters."
>
> This is the point I have been trying to make.
>
> A generalization bound needs to be "non-trivial"
>  - that it needs to show independence from some parameter on which one would have naively expected the bound to scale with.
>
> The whole point of generalization theory is to answer the fundamental question - "When are bigger models better?"
>
> This point is simply not getting made here - not at all readable from the given bound.
>
> Even if this is missing, the paper could have been acceptable had the experiment made the point of the theoretical bound being correlated to the true generalization error.
>
> With both these key points missing I am unable to change my scores.

---

> > ### Author Response · Authors · 2023-11-21
> > **Reponse to reviewer c9Ry**
> >
> > We thank the reviewer for his constructive feedback and for engaging in this interesting and stimulating discussion.
> >
> > In our opinion, analysing how the complexity of a model scales with the number of parameters is a question that is just as interesting as proving bounds that are invariant w.r.t. to a given parameter. We stress again that the results of Chen (2020) provide such an analyse: their bounds do not display independence w.r.t. to a parameter of their model, but carefully examine how strong the dependence is. As pointed out earlier, given a similar set of assumptions, our bounds are of the same nature. In the case where sampling is regular, i.e. we consider a sequence of sampling schemes with $K$ equidistant points, the discretization dependant quantities can be upper bounded by the quantities appearing in the continuous time analysis of our model (see Remark 3.5). This means that in this case, our bounds are length independent. This is close to the results of Marion (2023), who has obtained a similar independence when analysing neural ODEs and ResNets.
> >
> > We also wish to underline that our bounds taken as a whole are far from being trivial. In addition to upper bounding the Rademacher complexity of NCDEs, we have also given a rigorous theoretical framework to study them and a novel approximation bound. These bounds display dependence on the sampling gap, which is highly relevant for practical applications of NCDEs. All these results are absent from the literature and are, in our opinion, of high interest since NCDEs are an interesting model, with strong empirical results and a growing body of applications (see our Introduction).
> >
> > We encourage the reviewer to consider all these contributions when rating our work and thank him again for raising an interesting series of questions.

---

### Meta-Review · Area_Chair_A2vt · 2023-12-12

**Metareview:**

The manuscript analyzes the approximation and generalization errors of the neural controlled differential equations (NCDE).

**Justification For Why Not Higher Score:**

It is arguable whether the NCDE is a mainstream architecture, so that the impact of this analysis is unclear.

**Justification For Why Not Lower Score:**

The generalization bound for NCDE is a worthy contribution and the analysis is non-trivial.

---

### Decision · Program_Chairs · 2024-01-16

Accept (poster)